# Directed self-assembly of a xenogeneic vascularized endocrine pancreas for type 1 diabetes

Antonio Citro [1] ✉, Alessia Neroni[1,2], Cataldo Pignatelli[1], Francesco Campo [1,2], Martina Policardi[1], Matteo Monieri[1], Silvia Pellegrini[1], Erica Dugnani[1], Fabio Manenti[1], Maria Chiara Maffia[1], Libera Valla [1,3,4], Elisabeth Kemter [3,4,5], Ilaria Marzinotto [1], Cristina Olgasi[6], Alessia Cucci[6], Antonia Follenzi [6], Vito Lampasona [1], Eckhard Wolf [3,4,5] & Lorenzo Piemonti [1,2]

Intrahepatic islet transplantation is the standard cell therapy for β cell replacement. However, the shortage of organ donors and an unsatisfactory engraftment limit its application to a selected patients with type 1 diabetes. There is an urgent need to identify alternative strategies based on an unlimited source of insulin producing cells and innovative scaffolds to foster cell inter-action and integration to orchestrate physiological endocrine function. We previously proposed the use of decellularized lung as a scaffold for β cell replacement with the final goal of engineering a vascularized endocrine organ. Here, we prototyped this technology with the integration of neonatal porcine islet and healthy subject-derived blood outgrowth endothelial cells to engineer a xenogeneic vascularized endocrine pancreas. We validated ex vivo cell integration and function, its engraftment and performance in a preclinical model of diabetes. Results showed that this technology not only is able to foster neonatal pig islet maturation in vitro, but also to perform in vivo immediately upon transplantation and for over 18 weeks, compared to normal performance within 8 weeks in various state of the art preclinical models. Given the recent progress in donor pig genetic engineering, this technology may enable the assembly of immune-protected functional endocrine organs.

The primary treatment for patients with type 1 diabetes mellitus (T1D) consists of the combination of glucose monitoring coupled with daily exogenous insulin injections that, despite their effectiveness, do not prevent the challenges associated with daily compliance and T1D secondary complications[1]. In addition, the dynamic glucose/insulin homeostasis control mediated by the pancreatic islets of Langerhans in response to changes in blood glucose concentration is imperfectly simulated by periodic insulin injection[2]. Currently, another available

option is β cell replacement, which consists in islet/pancreas transplantation[3], but, at present, several limitations have been imposed by the shortage of organ donors and the immune reaction against the graft, despite the advances in immunosuppressive treatments[3].

The next step towards clinical application of an alternative treatment is the identification of sources of endocrine and vascular cells compatible with implantation in humans and their application as β cell

[1]San Raffaele Diabetes Research Institute, IRCCS San Raffaele Scientific Institute, Milan, Italy. [2]Università Vita-Salute San Raffaele, Milan, Italy. [3]Chair for Molecular Animal Breeding and Biotechnology, Gene Center and Department of Veterinary Sciences, LMU Munich, Munich, Germany. [4]Center for Innovative Medical Models (CiMM), LMU Munich, Oberschleißheim, Germany. [5]German Center for Diabetes Research (DZD), Neuherberg, Germany. [6]Department of Health Sciences, School of Medicine, University of Piemonte Orientale, Novara, Italy. ✉e-mail: citro.antonio@hsr.it

therapy based on scaffold generation to implement less immunogenic replacement strategies. The current organ donor supply will never meet the clinical demand, thus alternative islet sources of unlimited supply are being explored, such as stem cell-derived products and pig islet[4–6]. Pig is the favorite donor species for xeno-islets for several reasons: (1) porcine insulin is active in humans, (2) high fecundity and short generation time of pigs, (3) pigs can be maintained under designated pathogen-free conditions and, most importantly, (4) genetic engineering and gene editing tools have been adapted to pigs to overcome rejection mechanisms, improve islet function, and reduce the risk for zoonosis[7]. Furthermore, the isolation of neonatal pig islets (NPIs) is straightforward and can be scaled to therapeutic quantities[6,7]. However, compared with adult pig islets, the insulin content of NPIs is only 10–20%, and they require functional maturation to achieve physiological endocrine activity[8,9]. As of now, immature NPIs have been tested both in vitro and in vivo in the presence of supporting native or synthetic extracellular matrix (ECM) components to foster endocrine maturation without showing an immediate orchestrated mature function[10–14]. Growing evidence indicates that the microenvironment plays a crucial role in cell behavior, differentiation, and function[15]. ECM components, geometry, and stiffness can induce and promote intracellular biochemical signaling and are thus key players in modulating immature cell differentiation to matured ones[16–18]. We previously proposed the use of decellularized lung as a scaffold for β cell engraftment ex vivo and demonstrated that the seeding of the endocrine side of the pancreas in a dedicated vascular bed in combination with native ECM dramatically improved β cell survival, function, and in vivo performance[19]. Therefore, the use of decellularized organ ECM could positively impact the survival and maturation of the endocrine cells/clusters[15,16].

To this aim, this study shows a proof-of-feasibility of utilizing NPIs as the primary tissue source to engineer a native endocrine organ[20]. Moreover, the generation of autologous vascular beds should improve and foster endocrine engraftment offering a self-endothelial barrier. Here, we demonstrate the use of healthy subjects-derived blood outgrowth endothelial cells (BOECs) as an endothelial primary cell source to engineer a native organ vascular network[20]. Thus, we took advantage of our bioengineering platform, based on decellularized rat lung left lobe repopulated with endothelial cells through the organ native vascular access and endocrine cells through the airspace, to move a step forward to the clinical application by combining BOECs directly isolated from human peripheral blood and freshly isolated NPIs to bioengineer a xenogenic vascularized endocrine pancreas (VEP).

## Results

### BOEC characterization for VEPs assembly

To develop a functional pre-vascularized endocrine organ, we engineered scaffolds seeded with NPIs and healthy study participants BOECs. As preliminary step, we first isolated and characterized BOECs from healthy study participants as previously reported[21]. BOECs, at both early and late passages, preserved their endothelial phenotype as confirmed by strong positivity for endothelial markers by both immunofluorescence (CD31+/Von Willebrand+, VE-cadherin+ and VEGFR2+, Fig. 1A) and in flow cytometry (CD31, VE-cadherin and KDR positive cells, Fig. 1B, representative gate strategy in supplementary Fig 1) where, as expected, they also stained negative for CD34. Subsequently, we tested the BOECs' phenotype preservation and proliferation performance in the presence of modified media for vascularized endocrine organ generation[19] (VEP media). BOECs showed significantly improved proliferation over 24 and 48 h of culture in the presence of VEP media compared to the standard BOEC culture media (BCM) (Fig. 1C, ***WST1 test. VEP; $p = 0.0002$ for both times). Additionally, the BOECs' ability to form vascular structures was assessed by performing tube formation assay in the presence of BCM or VEP media combination. Since the VEP culture process is based on a

two-phase culture media protocol[19], the tube formation assay was adapted to mirror it. The VEP culture media combination showed a significant improvement in promoting BOEC tube formation compared to the BCM (Fig. 1D, E, tube formation assay - Mean Mesh size: BCM vs. VEP media combination $p = 0.0235$).

### Directed self-assembly of VEPs and functional assessment

NPIs, both iRFP transgenic and WT, were isolated from 1- to 6-day-old piglets, and the iRFP signal was confirmed, as previously described[22]. After 10 days of NPI in vitro maturation, isolated clusters were seeded in combination with BOECs in decellularized lung rat left lobes to bioengineer the VEP. First, we repopulated acellular lung matrices by seeding BOECs from both the pulmonary artery (PA) and pulmonary vein (PV). The day after, we repopulated the airspace with the combination of BOECs and NPIs through the trachea (T) (Fig. 2A). As already described, we cultured the resulting constructs for an additional 4 days in AM, followed by 2 days in MSM[19] (Fig. 2A). By the end of this maturation process, we observed a precise spatial organization in the bio-engineered organ of distinct tissue compartments, with a continuous human endothelial cell network and mature insulin positive NPIs integrated in the newly formed BOEC vascular bed (Fig. 2B, C). Thus, to investigate whether the newly formed vasculature architecture would allow direct perfusion of the VEP structure, we perfused the PA with 0.2-μm microspheres at the end of maturation process. Confocal microscopy of the VEP section showed the retention of the blue microspheres in the engineered vascular bed, suggesting that the newly formed human BOEC endothelium is preserved and distributes the perifusate in a dynamic 3D organization (Fig. 2C). Taken together, these findings suggested that the VEP provided direct perfusion of seeded NPIs due to functional integration of the endocrine clusters in the bio-engineered vascular architecture.

We further investigated the endocrine maturation of NPIs in VEP by evaluating insulin, glucagon, and somatostatin mRNA and protein expression levels (Fig. 2D–G; Supplementary Figs. 2). Batch matched NPIs in standard culture or in VEP were analyzed at day 1 and 7 (Fig. 2D–G). At protein level, day 7 VEPs showed significantly increased staining for insulin with a preserved glucagon and somatostatin positivity compared to day 1 VEPs (Fig. 2E–G). Concordantly, we observed a 3.5-, 1.1- and 1.9-fold increased insulin, glucagon, and somatostatin mRNA levels in day 7 compared to day 1 VEPs (Supplementary Fig. 2). Additionally, higher levels of insulin and glucagon fluorescence were evident in NPIs seeded in VEP compared to batch-matched NPIs in standard cultures at day 1 and 7 (Fig. 2E–G), confirming that the presence of the vascularized ECM is an inducer of maturation.

We also investigated VEPs endocrine secreting performance. NPIs were evaluated with a dynamic insulin secretion test in VEPs compared to batch-matched NPIs cultured in standard conditions in petri dishes at day 7 (Fig. 3A–C). When exposed to high glucose concentration (20 mM), VEPs demonstrated a physiologic insulin release, while NPIs maintained in standard culture conditions were not able to show an orchestrated insulin response. Moreover, VEPs released significantly higher insulin levels as shown by the point-by-point analysis (VEP vs. NPIs min 2 $p = 0.026$; min 3 $p < 0.0001$; min 4 $p = 0.0002$; min 5 $p = 0.0155$; min 7 $p = 0.0085$). The peak of insulin release of matured VEPs during the first phase of secretion was significantly higher compared to NPIs from standard culture conditions (min 3: VEP fold change $2.02 \pm 0.43$ vs. NPIs fold change $0.89 \pm 0.03$; $p < 0.0001$), and a significant difference was also observed in the second phase of insulin secretion (min 7: VEP $1.27 \pm 0.3$ vs. NPIs $0.74 \pm 0.32$ fold-change; $p = 0.0085$, Fig. 3A). The AUC of insulin release of VEPs during the insulin response was significantly higher in both phases (AUC VEP first phase: $3.43 \pm 0.9$; second phase: $2.37 \pm 0.5$) compared to the NPIs in standard culture conditions (AUC first phase NPI $1.6 \pm 0.2$, $p < 0.0001$; second phase: $1.43 \pm 0.5$, $p = 0.0031$, Fig. 3B). Additionally, batch-matched VEPs compared at 7 or 14 days of culture showed preserved

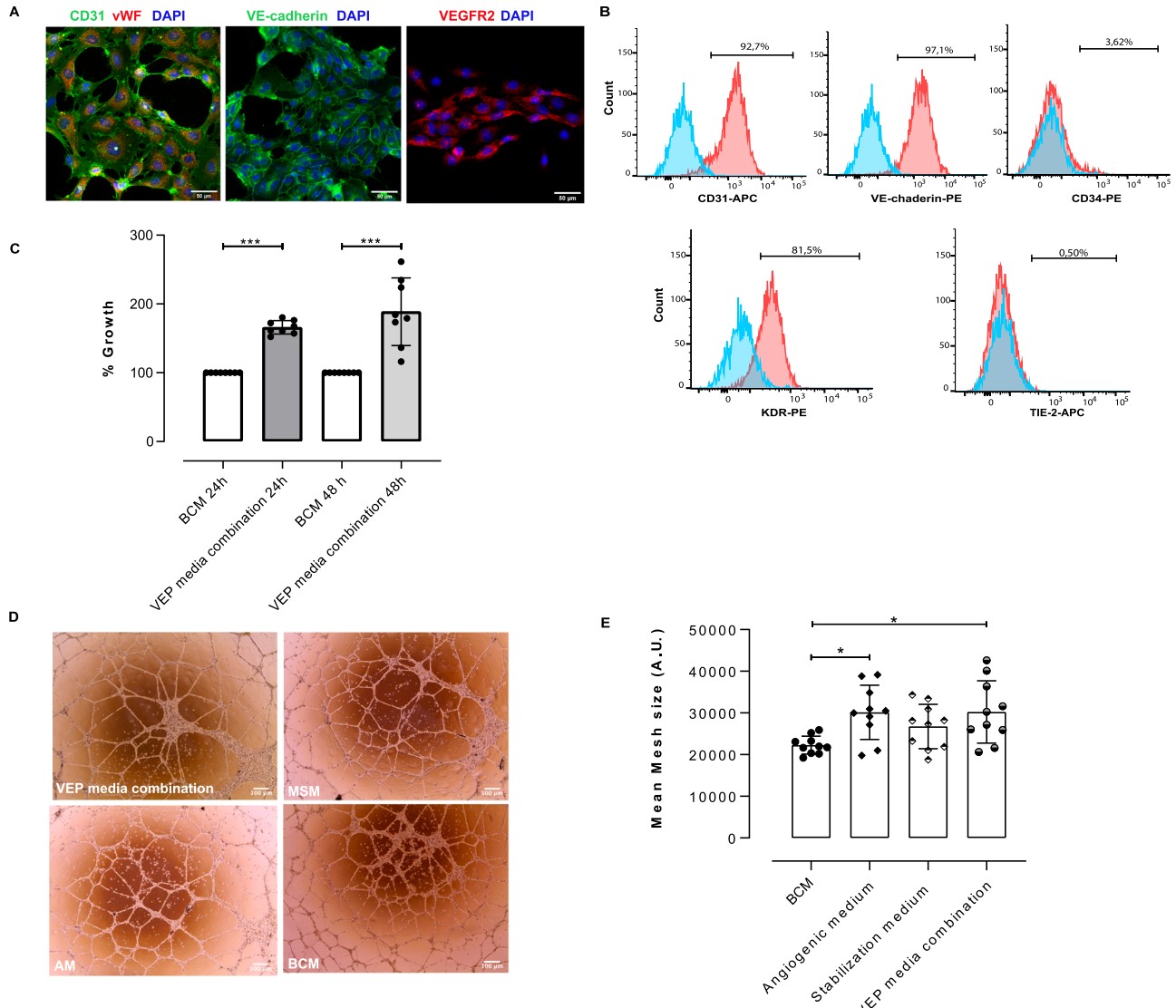

**Fig. 1 | BOEC functional characterization for VEPs assembly.**
**A** Immunofluorescence image of BOEC endothelial phenotyping CD31 (green) and Von Willebrand factor (vWF-red), VE-cadherin (green), vascular endothelial growth factor receptor 2 (VEGF-R2-red) and DAPI (blue). Scale bar in μm. Two independent experiments; the representative images were shown; **B** Flow cytometry BOECs phenotyping for CD31, VE-cadherin, CD34 and KDR (blue and red: histogram for isotype control and positive cells respectively); **C** WST-1 assay for BOEC proliferation in the presence of standard BOEC culture medium (BCM) or VEP media combination, ($n = 8$ biological replicates) for 24 or 48 h. Values presented as mean ± SD. BCM *vs*. VEP media combination 24 h ***$p = 0.0002$; BCM *vs*. VEP media

combination 48 h ***$p = 0.0002$; Mann-Whitney U-test **D**, **E** BOECs tube formation assay: **D** Morphology of BOECs tube formation assay in the presence of VEP media, MSM Modified Stabilization Medium, AM Angiogenic Medium, and BCM magnification 4×. Two independent experiments; the representative images are shown. **E** Analysis of the formed vascular mesh for vascular density analysis in the presence of BCM, AM, MSM and VEP media combination, ($n = 10$ biological replicates), values presented as mean ± SD (BCM *vs*. Angiogenic medium *$p = 0.0166$; BCM *vs*. VEP media combination *$p = 0.0235$), one way ANOVA with Dunn's correction. Source data are provided as a Source Data file.

biphasic insulin secretion kinetics coupled with a trend in improving the overall insulin production over time (Fig. 3C). To further dissect the role of the VEP engineering process on the functional maturation of NPIs, we investigated the role of endothelial cells by evaluating the endocrine performance at day 7 of VEP generated in the presence or absence of BOEC (VEP$^{-BOEC}$). NPIs in the VEP platform, in the absence of the engineered vascular network, showed a significantly reduced performance compared to the fully matured VEP as shown by the point-by-point analysis (VEPs *vs* VEPs$^{-BOEC}$ ** min 3 $p = 0.009$ and * min 4 $p = 0.0157$, ANOVA for multiple comparisons, supplementary Fig. 3). This suggests that endothelial cells play a relevant role in the maturation process of NPIs within the VEP.

Furthermore, to visualize the VEP's ability to preserve the β cell mass, we tracked NPIs death during VEP maturation by measuring

miR-375. MiR-375 is a highly expressed microRNA in pancreatic β cells and previous studies suggested that it is a suitable biomarker for real-time detection of β cell death[23–26]. Based on the estimated number of miR-375 copies per cell and the miR-375 copies measured in the VEP supernatants at each time-point, we estimated the percentage of β cell death during the VEP maturation process. To rule out bias related to miR-375's contribution in the VEP engineering process, we confirmed the absence of miR-375 release over 7 days by a VEP containing exclusively BOECs (VEP-NPIs, Fig. 3D−blue line). The kinetics of miR-375 release during the 7-day VEP maturation process is reported in Fig. 3D. The normalization by miR-375 copies per cell estimated a very low rate of β cell death, ranging between 0 and 5%, during the 7-day maturation process (Fig. 3E).

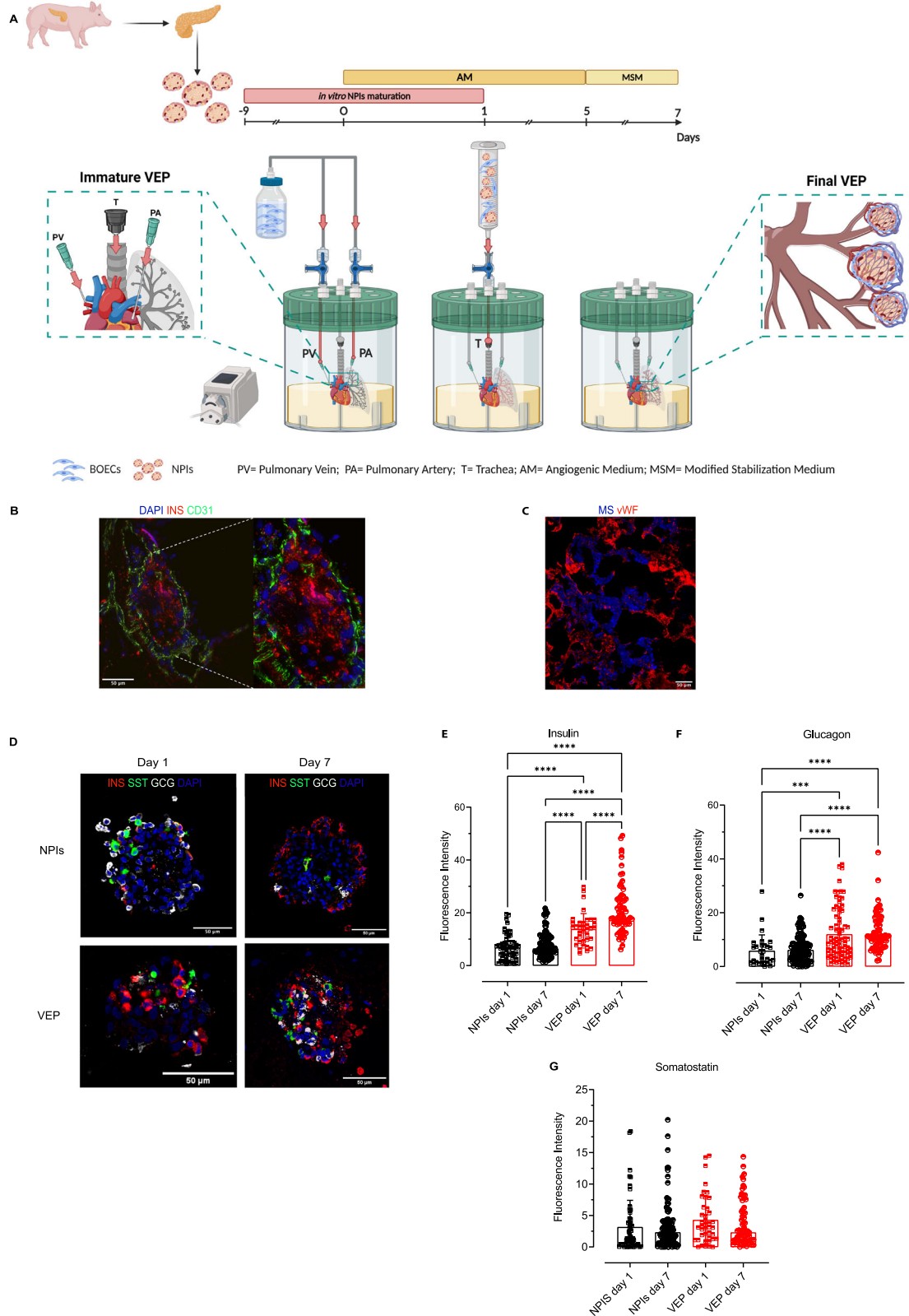

## VEP performance after transplantation into diabetic mice

Mature VEPs were tested in vivo in a preclinical model of severe diabetes in immune-compromised NSG mice (Figs. 4 and 5). VEPs performance were compared to batch-matched NPIs transplanted in a device-less, pre-vascularized subcutaneous pouch (DL-NPI), under the kidney capsule (KC-NPIs), or in the liver through the portal vein (LV-NPIs) (Fig. 4A). As previously reported[19], the VEP was segmented longitudinally in two different portions to allow organ integration within the unmodified subcutaneous implantation area (Fig. 4A). Graft function was monitored by measuring glucose levels over 9 weeks for DL-NPIs and LV-NPIs, and over 18 weeks for KC-NPIs and VEPs experimental groups. Non-fasting glycemia of VEP implanted mice was significantly lower compared to the other groups (VEPs $vs.$ DL-NPIs $p = 4.98e\text{-}39$; VEPs $vs.$ KC-NPIs $p = 3.85e\text{-}29$ and VEPs $vs.$ LV-NPIs $p = 2.01e\text{-}25$; general linear model, repeated measures corrected by Bonferroni, Fig. 4B). Of note, 37% of mice carrying VEPs never required

**Fig. 2 | VEP assembly and endocrine maturation assessment. A** Schematic drawing of customized bioreactor and steps in VEP assembly: from NPIs isolation to VEP maturation (AM Angiogenic Medium, MSM Modified Stabilization Medium, PA Pulmonary Artery, PV pulmonary Vein, and T Trachea). Created with Biorender. **B** Representative immunofluorescence of 7 days mature VEP after culture in bioreactor system. (Insulin−INS: red, CD31: green, DAPI−blue) Scale bars in µm. Two independent experiments, the representative images are shown. **C** Fluorescence microsphere perfusion. 0.2-µm microspheres (MS, blue,) present and trapped in newly formed VEP vasculature (human vWF, red). Scale bars in µm. Two independent experiments, the representative images is shown. **D** Immunofluorescence comparison of 1 (left) and 7 days (right) batch matched NPIs alone or in VEP. (Insulin−INS: red, somatostatin STT: green, glucagon−GCG: white, DAPI: blue) Scale bars in µm. **E**−**G** Insulin, glucagon, and somatostatin immuno-fluorescence intensity

quantification on VEP (day 1 and day 7) and batch matched NPI (day 1 and day 7), $n$ represents islet from 4 pancreata; one way ANOVA for multiple comparison corrected by Tukey, values presented as mean ± SD: **E** Insulin: NPIs day 1 ($n = 56$ islets), day 7 ($n = 114$ islets), VEP day 1 ($n = 36$ islets), day 7 ($n = 71$ islets); VEP day7 $vs$. NPIs day1 ****$p = 2.23e$-13; VEP day 7 $vs$. NPIs day 7 ****$p = 2.23e$-13; VEP day 1 $vs$. NPIs day 1**** $p = 4.74e$-06; VEP day 1 $vs$. NPIs day 7 ****$p = 1.98e$-05; VEP day 1 $vs$. VEP day 7 ****$p = 2,23e$-07. **F** Glucagon: NPIs day 1 ($n = 29$ islets), day 7 ($n = 111$ islets), VEP day 1 ($n = 77$ islets), VEP day 7 ($n = 70$ islets); VEP day 7 $vs$. NPIs day 7 ****$p = 4.5e$-07; VEP day 7 $vs$. NPIs day 1 ****$p = 6.65e$-05; VEP day 1 $vs$. NPIs day 7 ****p9.59e-06; VEP day 1 $vs$. NPIs day 1 ***$p = 4.72e$-04. **G** Somatostatin: NPIs day 1 ($n = 56$ islets), day 7 ($n = 110$ islets), VEP NPIs day 1 ($n = 41$ islets), VEP NPIs day 7 ($n = 69$ islets). Source data are provided as a Source Data file.

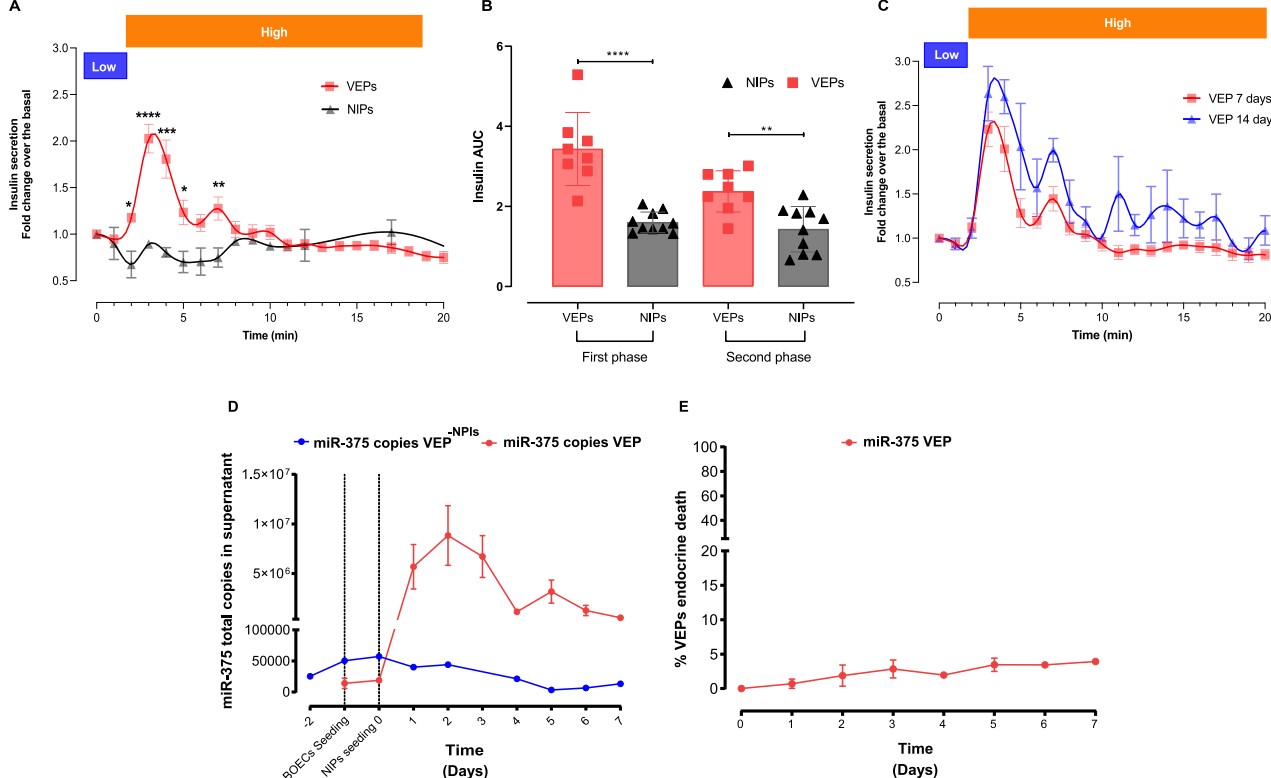

**Fig. 3 | Ex vivo functional evaluation of VEPs. A** Insulin secretion test (IST) of VEPs (red line, $n = 8$ scaffolds) $vs$. NPIs (black line, $n = 10$, NPIs preparations) after 7 days of culture in the presence of low (2 mM) and high (20 mM) glucose. Values are expressed as fold change over the basal. (VEP $vs$. NPIs min 2 *$p = 0.026$; min 3 ****$p < 0.0001$; min 4 ***$p = 0.0002$; min 5 *$p = 0.0155$; min 7 **$p = 0.0085$, Mann-Whitney U test) Values presented as mean ± SEM. **B** Area Under the Curve (AUC) analysis of IST. Analysis of VEPs ($n = 8$ scaffolds) and NPIs ($n = 10$ NPIs preparations) performance within first and second insulin secretion phase after glucose stimulus. AUC VEP first phase: 3.43 ± 0.9; second phase: 2.37 ± 0.5; AUC first phase NPI

1.6 ± 0.2; second phase: 1.43 ± 0.5; ****VEPs Vs NPIs first phase $p < 0.0001$; VEPs Vs NPIs second phase: $p = 0.0031$. Mann-Whitney U test, Values presented as mean ± SEM. **C** IST of VEPs ($n = 5$ scaffolds) $vs$. NPIs ($n = 2$ NPIs preparations) after 7 or 14 days of culture, respectively. Values presented as mean ± SEM. **D** miR-375 kinetic release profile (absolute copies released in supernatant) during 7 days of culture of mature VEPs (red, $n = 6$ scaffolds) and VEPs−NPIs (blue, $n = 1$ scaffold). Values presented as mean ± SEM. **E** Percentage of total β cell death during 7 days VEPs culture ($n = 6$ scaffolds). Values presented as mean ± SD. Source data are provided as a Source Data file.

insulin pellet due to immediate graft function after implantation, showing a rapid functional maturation of endocrine cells. Conversely, all animals in the other experimental groups showed suboptimal glucose control with the need for insulin pellets ($p = 0.025$, chi-square test). Within the follow up, normoglycemia was restored in 20 out of 20 (100%) of VEP-implanted mice, while normoglycemia was achieved in only 1 out of 8 (12.5%), 4 out of 9 (44.4%), and 1 out of 4 (25%) of DL-NPIs, KC-NPIs, and LV-NPIs transplanted mice, respectively. The median time for achieving normoglycemia was 3 ± 0.6 days for VEPs, 65 ± 5.55 days for KC-NPI, and ≥ 60 days for DL-NPIs and LV-NPIs ($p < 0.0001$, log rank analysis) (Fig. 4C). An oral glucose tolerance test was performed in all groups 9 weeks after transplantation (Fig. 4D). After glucose loading, VEP mice showed a significantly lower blood

glucose concentration compared to DL-NPI mice (AUC VEPs vs DL-NPIs $p = 0.0463$), confirming the superior function of the engrafted scaffold (Fig. 4D, E).

To track the persistence of endocrine grafted tissues using non-invasive imaging, NPIs obtained from transgenic pigs expressing near-infrared fluorescent protein (iRFP) 720 were used in some in vivo experiments. As previously reported[22], we monitored the fate of NPI xenotransplants in mice by fluorescence imaging using an IVIS® SpectrumCT bioluminescent in vivo imaging system. Consistent with the functional data, iRFP signal was present in the early days but was no longer detectable at 9 weeks after implantation in the DL group (Supplementary Fig. 4); conversely, the iRFP signal was detectable in all animals transplanted with VEPs and in normoglycemic KC-NPI and

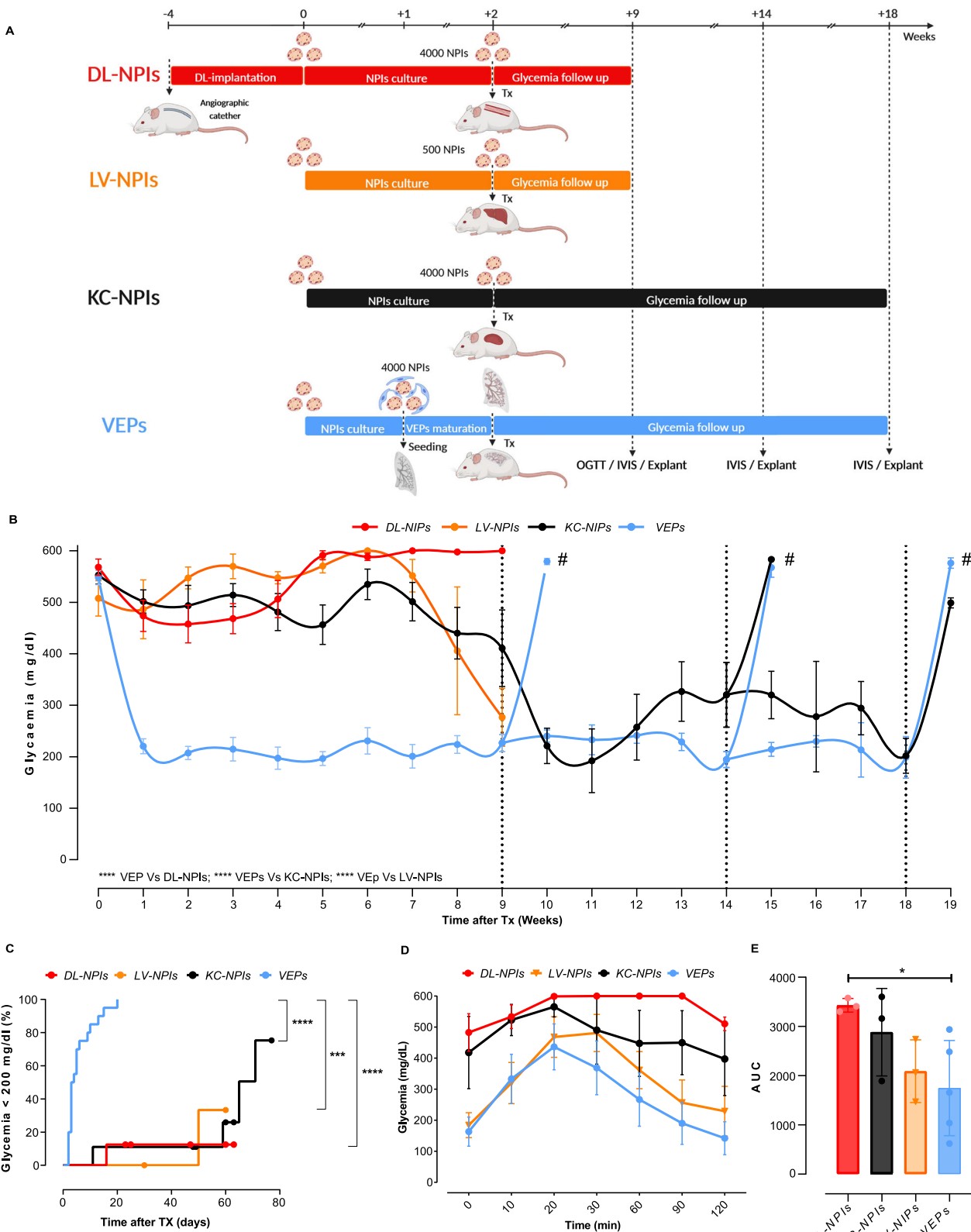

LV-NPI recipients (Fig. 5A). At 9, 14 or 18 weeks of follow up, the VEPs were explanted to confirm the graft's contribution to the mice's glucose control. As expected, immediately after VEP explant, the mice reverted to a diabetic status and displayed hyperglycemia for at least one additional week (Fig. 4B). Moreover, iRFP signal was detected in the graft-bearing explanted tissue at 9, 14 and 18 weeks (Fig. 5B). As reported above, the presence of endothelial cells appears relevant in the NPI maturation process in vitro, since the seeding of

NPIs in the VEP in the absence of BOEC determines a delayed functional maturation (Supplementary Fig. 3). To dissect the specific role of BOECs in the engraftment process, regardless of their action on endocrine maturation, VEPs were engineered in the absence of BOECs (VEP-BOEC) but in the presence of NPIs fully matured in culture for 21 days[8]. VEP-BOEC were transplanted in diabetic NSG mice (Supplementary Fig. 5A). Results showed that the absence of BOEC delayed the gain of normoglycemia by four weeks (Supplementary Fig. 5B),

**Fig. 4 | Long-term in vivo performance of VEPs. A** Schematic representation of in vivo experimental protocol. Created with Biorender. **B** Weekly not fasting glycemia profile comparison between DL-NPIs (red, *n* = 8 implantations), LV-NPIs (yellow, *n* = 4 implantations), VEPs (blue, *n* = 20 implantations) and KC-NPIs (black, *n* = 10 implantations) for 9, 14, or 18 weeks follow up (# represents graft explants: 9 weeks VEP n = 4 explants; 14 weeks KC-NPIs=1explants; VEPs *n* = 4 explants; 18 weeks KC-NPIs *n* = 3 explants, VEPs n = 2 explants). Values presented as Mean ± SEM. (**** VEPs *vs.* DL-NPIs *p* = 4.98e-39; VEPs *vs.* KC-NPIs *p* = 3.85e-29, and VEPs *vs.* LV-NPIs *p* = 2.01e-25; general linear model for repeated measures corrected by Bonferroni). **C** Kaplan Mayer analysis of the percentage of mice reaching

normoglycaemia (≤200 mg/dl). Differences between VEPs, DL-NPIs, KC-NPIs and LV-NPIs were estimated by Log Rank test adjusted for multiple comparisons with Benjamini & Hochberg method. (**** VEPs *vs.* DL-NPIs *p* = 1.2e-05; VEPs *vs.* KC-NPIs *p* = 1.2e-05; *** VEPs *vs.* LV-NPIs *p* = 0.00074). **D** VEPs (*n* = 5 mice), DL-NPIs (*n* = 3 mice), KC-NPIs (*n* = 3 mice) and LV-NPIs (*n* = 3 mice) OGTT performance 9 weeks after transplantation. Values presented as mean ± SEM. **E** OGTT AUC over 120 min at 9 weeks after implantation (*n* = 5 in VEPs mice, *n* = 3 in DL-NPIs mice, *n* = 3 in KC-NPIs mice and *n* = 3 LV-NPIs mice). * *p* = 0.0463; one way Anova for multiple comparison corrected for Dunn's. Values presented as mean ± SD. Source data are provided as a Source Data file.

**A**

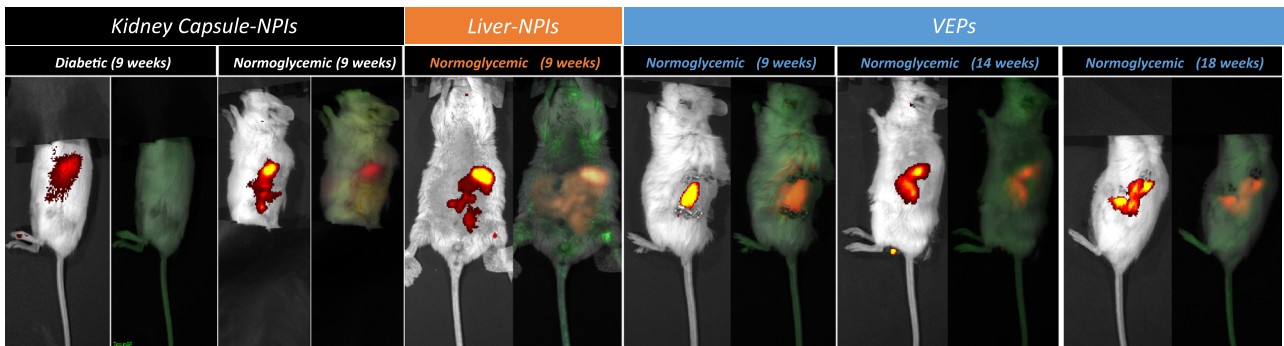

**B**

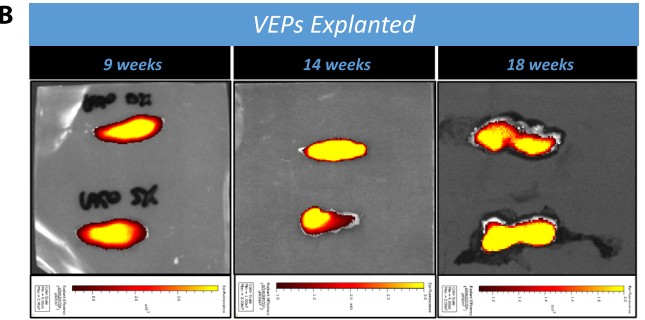

**C**

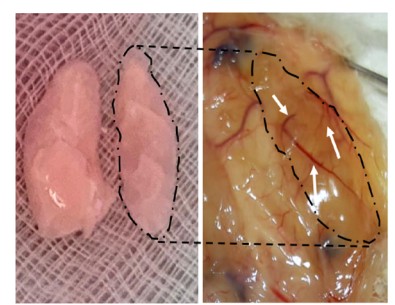

**D**

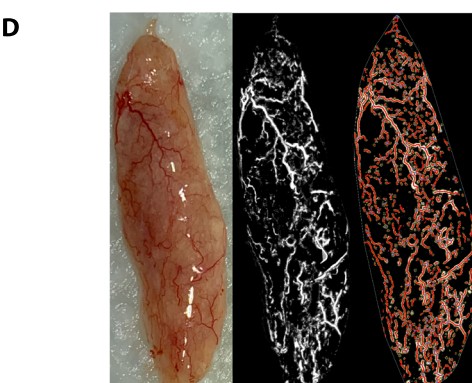

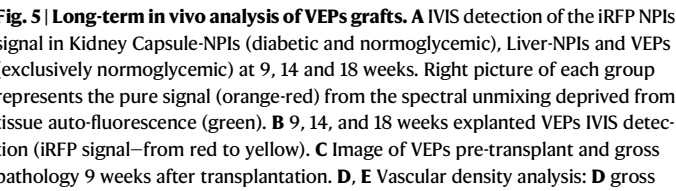

**E**

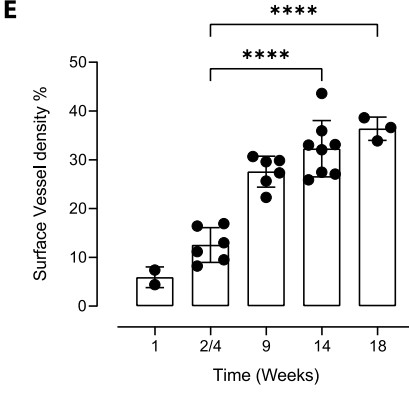

**Fig. 5 | Long-term in vivo analysis of VEPs grafts. A** IVIS detection of the iRFP NPIs signal in Kidney Capsule-NPIs (diabetic and normoglycemic), Liver-NPIs and VEPs (exclusively normoglycemic) at 9, 14 and 18 weeks. Right picture of each group represents the pure signal (orange-red) from the spectral unmixing deprived from tissue auto-fluorescence (green). **B** 9, 14, and 18 weeks explanted VEPs IVIS detection (iRFP signal−from red to yellow). **C** Image of VEPs pre-transplant and gross pathology 9 weeks after transplantation. **D**, **E** Vascular density analysis: **D** gross

pathology of 9 weeks VEP explants (left panel) and its computational software analysis of vascular VEP surface (middle-right panel), **E** percentage of VEP surface vessel density 1 (*n* = 2 explants), 2/4 (*n* = 6 explants), 9 (*n* = 6 explants), 14 (*n* = 7 explants) and 18 (*n* = 3 explants) weeks after implantation. Values are summarized as mean ± SD. One way anova for multiple comparisons corrected by Tukey (*** 2/4 *vs.*14 *p* = 3.72e-07: **** 2/4 *vs.*18 *p* = 1.3e-06). Source data are provided as a Source Data file.

suggesting that the presence of endothelial cells represents a key factor in enabling NPIs engraftment.

Gross pathology evaluation of explanted VEPs (Fig. 5C, D) showed a complete integration of the engineered structure in the subcutaneous space. The VEP structure was easily recognizable, and blood perfused vascular structures of different caliber were identified within the VEP architecture (Fig. 5C, white arrows). We performed a vascular density analysis of VEPs explanted at 1, 2, 4, 9, 14, and 18 weeks. Capillaries and vessel nodes were tracked and measured using the unbiased ANGIO TOOL 64 version 0.6a software[27] (Fig. 5D, E). Results showed a significant improvement of the vascularization over time, with a surface vessel density of $12.53 \pm 3.58\%$ at 2/4 weeks compared to $32.26 \pm 5.78$ and $36.37 \pm 2.37$ at 14 weeks ($p < 0.0001$) and 18 weeks ($p < 0.0001$), respectively (ANOVA for multiple comparisons). Additionally, immunofluorescence analysis of VEPs at 9 weeks revealed the presence of both human and murine endothelium within the implanted organ vascular architecture (Supplementary Fig. 6). TEM analysis performed on the explanted graft at 14 weeks showed the presence of capillaries filled with red blood cells and the presence of nerve structures (confirmed also by immunofluorescence analysis) in the proximity of granulated endocrine cells within the VEP edges (Supplementary Fig. 7A–C). Although transplantations were performed in immunocompromised NSG mice, we also evaluated murine residual neutrophils and monocytes/macrophages by immunofluorescence for CD11b and LY6G. As expected, no infiltrating immune cells were identified in VEPs 9 weeks after transplantation (Supplementary Fig. 8).

## Discussion

Recent advances in endocrine pancreas bioengineering have showed the potential application of decellularized organs as native scaffold to safely harbor endocrine cells to control their engraftment[28–30]. In this context, we previously demonstrated the feasibility of an organ repurposing strategy by engineering a decellularized lung in a vascularized islet organ to restore normoglycemia in a preclinical model of diabetes[19]. Starting from this first attempt, we prototyped here an upgraded version that matches relevant criteria for future clinical application, demonstrating that the decellularized left lung lobe offers valuable support for integrating immature NPIs and BOECs. To engineer a clinically compatible vascular VEP compartment, we used BOEC as a source of endothelial cells. As first step, we assessed the compatibility of BOEC survival and function with VEP media for the bioengineering process, demonstrating the preservation of their phenotype and functional status. Although isolation efficiency can be impacted by the donor related colony formation capacity, BOEC may offer a number of important advantages over other endothelial cell sources. BOECs can be easily isolated from healthy and diabetic patients[20,21,31,32]. Furthermore, BOECs have a high proliferative capacity and are able to promote neovascularization in vivo in immunodeficient mice[33] suggesting that they can be directly involved in vessel formation. Moreover, BOECs have already been proposed for the treatment of hemophilia and as a tool in gene therapy approaches[21,34]. Finally, BOECs can release therapeutic agents straight into the blood stream due to their direct connection to the vascular system. The use of these cells in VEPs will increase the biocompatibility of the generated scaffold as the vascular compartment can be composed of autologous endothelial cells.

In the second step, we redesigned the VEP endocrine compartment by introducing immature NPIs as a valuable, unlimited source of insulin producing cells with a potential future in clinical translation. Indeed, pigs have a very similar embryonic development, anatomy, and physiological blood glucose level with that of humans. The similarities between humans and pigs, together with the feasibility of pig genome engineering and the demonstration that NPIs transplantation in adult pigs and non-human primates is able to reverse diabetes[35–37],

makes the use of NPIs an intriguing endocrine cell source to engineer new technologies in the β cell replacement field. After isolation, NPIs are immature and require, dependently by various pig sources, an in vitro dedicated culture protocol and/or in vivo step to mature into glucose-sensing endocrine producing cells[8], as evidenced by the late hyperglycemia reversal after transplantation in preclinical models of T1D[38–40]. With this in mind, we challenged our technology with a committed, but immature source of unlimited endocrine cells evaluating the ability of VEP platform to allow not only the ex vivo NPIs vascular integration within BOECs architecture, but also their functional maturation. To confirm the endocrine maturation, we first investigated the insulin, glucagon, and somatostatin expression at mRNA and protein levels. We confirmed that immature NPIs may improve their functional status over the culture time. More interestingly, we discovered that, compared to batch-matched control NPIs, the vascularized ECM significantly boosted NPIs endocrine maturation within the VEP. Next, we challenged mature VEP and batch-matched NPIs with an insulin secretion test, and we found an improved insulin production and mature physiological insulin secretion kinetics both at 7 and 14 days of VEP maturation. Additionally, NPIs in the VEP platform in the absence of the engineered vascular network, showed a significantly reduced performance compared to the fully matured VEP. This finding indicates not only a precise glucose triggered insulin release rather than stress-induced insulin dumping by damaged islets, but also an improved insulin secretion over time, suggesting a strong contribution of the vascularized ECM structure in improving NPI endocrine secreting activity.

A critical challenge in the endocrine ex vivo device manipulation is the survival of the engrafted β cell mass. To make the VEP's ability to preserve the β cell mass evident, we tracked NPI death during VEP maturation by measuring miR-375. MiR-375 was suggested as a reliable surrogate marker for β cell death[23,41] as it is the most abundant miRNA in pancreatic islets, with an important role in maintaining the β cell mass[42,43]. Given the stability of miRNAs in the blood[44], miR-375 has been proposed as a biomarker for tracking β cell death[45,46]. While its clinical application is presently limited by its releases from different cell types[46–50], we took advantage of the lack of miR-375 expression by BOECs to specifically track the VEP's ability to preserve endocrine mass during the ex vivo maturation process, as VEPs are exclusively based on BOECs and NPIs. Thus, we measured the miR-375 release and quantified the β cell death during the VEP maturation demonstrating the ability of the VEP technology to preserve the NPI survival.

We also tested VEP function in immunodeficient NSG diabetic mice. We compared the performance of NPIs implanted within VEP with that of NPIs transplanted in other standard sites (e.g., the avascular kidney capsule, the pre-vascularized subcutaneous space (device-less), and the well-vascularized liver). When compared to all other groups, the bio-fabricated VEPs showed significantly better performance, exhibiting immediate function after subcutaneous implantation and the preservation of the normoglycemic function till the end of the follow-up (up to 18 weeks). Despite recent reports on device-less strategy to efficiently restore normoglycemia in preclinical models[51,52] of murine and human islet transplantation, NPIs transplanted using pre-vascularized subcutaneous space approach failed to reverse diabetes in immunodeficient NSG diabetic mice in our hands. This may be explained by the high metabolic activity needed to achieve the functional maturation of the NPIs that usually requires 6–12 weeks in vivo, as well as by the low oxygenation of the subcutaneous site. The failure of the device-less strategy enhances even more the VEP value since it has always guaranteed a very early functional success in the subcutaneous site. To foster this concept, we also evaluated the absence of the pre-vascularization strategy on VEP engraftment time. Thus, we engineered VEPs in the presence of matured NPIs but in the absence of BOECs. Results showed that the absence of BOEC delayed the gaining of normoglycemia by four weeks, suggesting that the

presence of endothelial cells represents a key factor in enabling the NPIs' engraftment in VEP. In line with prior reports[40,53], NPIs transplantation under the kidney capsule was able to reverse diabetes within 8 weeks after implantation, a significant delay in comparison with VEP. Here, we also report for the first time the intrahepatic NPIs transplantation. We faced a limitation with the NPI implantable mass that was significantly reduced compared to the other sites. Liver site showed a trend in restoring normoglycemia similar to the kidney capsule, even using eight times fewer NPIs, confirming the concept that the timing of β cell function after implantation is mainly dependent on the generation of vascular connection and partially independent of the implanted mass. All these results confirmed that, once matured, ex vivo vascularized engrafted NPIs may perform faster and better compared to batch-matched clusters, both matured or not, implanted in different sites or in the absence of a pre-vascularized strategy.

To confirm the role of the preservation of the engrafted mass in defining the perimeter of VEP endocrine function we transplanted iRFP-NPIs into all the implantation sites and tracked the signal at the end of the follow-up. As expected, all VEPs and only the normoglycemic KC-NPIs and LV-NPIs were able to show preserved iRFP endocrine mass. This was additionally confirmed by the VEP explants that caused the restoration of the hyperglycemic condition in the NSG recipients, confirming its role in the assessment of the endocrine function.

As shown by these results and other publications, decellularized ECM may induce neurotrophic effects[54] and the success of β cell replacement strategy is strongly dependent on the vascular connection[19,52,55,56] and innervation[57,58] between host and implanted graft; thus, we assessed the level of vascular and nerve integration of implanted VEPs during the follow-up. We observed a significant improvement of the VEP surface vascularization after the implantation, which was characterized by a progressive integration of the vascular structure from host to the recipient and the presence of nerves in the VEP surface edges after 14 weeks that was not observed in early time point, which is likely the reason why the VEP is able to guarantee a rapid achievement of normoglycemia and its maintenance over time. Although we did not observed fenestration of VEP vessels in TEM acquisitions, it is known that in tissue remodeling after islet transplantation the presence of fenestrated endothelium is less mandatory for islet function. In fact, is described that endothelial cells lining is disrupted during islet isolation, and re-endothelialization after transplantation is a slow and incomplete process, despite the glucose metabolism normalization in the recipient[59]. In this context, although vessel fenestration will represent an additional key point to characterize the VEP nutrient exchange, we assume, based on early glycaemia normalization and long-term VEP function, that a hormonal exchange between endocrine cells and blood is functionally taking place.

In conclusion, the VEP platform demonstrated to be flexible and able to be engineered not only with primary mature islets but also with immature endocrine clusters, of which it is able to boost their endocrine maturation and secretory function, both in vitro and in vivo, showing immediate function upon transplantation compared to controls. Of note, given the recent clinical application of stem cell-derived β cells[60] or human islets[61,62] housed in devices for transplantation, we believe that stringent criteria such as biocompatible environment, based on inert native ECM, and scaffold with functional architecture to promote ex vivo cell engraftment, interaction, and integration are crucial to generate the next level of endocrine devices for cell therapy in T1D. To our knowledge, by taking advantage of the native lung compartmentalized architecture, VEP is the only organ platform that offers the unique option to perform an ex vivo engraftment of endocrine cluster and autologous endothelial cells in a controlled environment. Additionally, since the VEP is based on native ECM, it offers the option to specifically modify the matrix component of the different compartment to foster cell integration or to better functionalize the final product. In this scenario, here we designed the VEP to match those criteria for the generation of a flexible device. Although further improvement needs to be made on the clinical grade VEP manufacturing, given recent progress in porcine organ transplantation[63] and gene editing for the generation of stealth grafts[63,64], VEP technology may enable the assembly of functional endocrine organs based on gene edited pig organs/NPIs and autologous BOECs from T1D patients to fully escape immunoreaction against the graft.

## Methods

### Animals
Experiments involving rats and mice were performed under protocols approved and monitored by the Animal Care and Use Committee of San Raffaele Scientific Institute. 6–8 weeks old male Lewis rats (175–200 g; Charles River Laboratories) were used as scaffold donors. Rats were kept in 12 h/12 h light/dark cycles, at 21.5 ± 1.5 °C temperature and 55% ± 15 humidity. 6-8 weeks NSG female immune-compromised mice (22–25 g; Charles River Laboratories, Calco, Italy) were used as transplant recipients. Mice were kept in 12 h/12 h light/dark cycles, at 21.5 ± 1.5 °C temperature and 55% ± 15 humidity. NPIs were isolated from donor pancreas of 1–7 days German Landrace for WT (66% males and 34% females) or CAG-iRFP (44% males and 56% females) transgenic piglets, latter ones expressing ubiquitously the near-infrared fluorescent protein iRFP720[22]. Piglets were housed in a farrowing pen together with the mother, and a heated nest were offered to the piglets. Those procedures were approved by district governments of Upper Bavaria, Germany and were conducted in accordance with the German Animal Welfare Act.

### Preparation of acellular lung scaffolds
Cadaveric rat lungs were isolated and perfused to allow their decellularization[65,66]. Cadaveric lungs were explanted from male Lewis rats (175–200 g, Charles River Laboratories) after systemic heparinization. The pulmonary artery (PA) was cannulated with an 18 G cannula (McMaster-Carr), the pulmonary veins (PV) were cannulated through the left atrial appendage using a miniball cannula with tip basket (1.9 mm ID) (Harvard Apparatus), and the aorta was ligated. The trachea was cannulated with a 16 G cannula (McMaster-Carr) and the right main bronchus was ligated. Subsequently, the right upper, middle, and lower lobes were tied off and removed. Decellularization was performed by perfusing the PA (constant pressure, 40 mm Hg) sequentially with heparinized (10 units/ml) phosphate-buffered saline (PBS, 10 min), 0.1% sodium dodecyl sulfate (SDS) in deionized water (2 h), deionized water (20 min), and 1% Triton X-100 in deionized water (20 min). The resulting scaffolds were washed with PBS supplemented with 1% antibiotics and antimycotics for 72 h to remove residual detergent and cellular debris. All reagents were sourced from Sigma-Aldrich. Scaffolds showing evidence of infection, vascular air trapping or vascular leakage were excluded from the study.

### Blood outgrowing endothelial cells
Blood sampling from healthy donors for BOEC isolation was performed at A.O.U. Maggiore della Carità, Novara, Italy or at San Raffaele Hospital, Milan, Italy. The procedure was approved by the Ethics Committee "Comitato Etico Interaziendale A.O.U. Maggiore della Carità" (Protocol 875/CE, Study n. CE 183/20, approval 07/28/2020) and by the ethical committee of the San Raffaele Hospital (Protocol hVIO, approval on 04/11/2019) and all the donors provided written, informed consent. BOECs were isolated from starting from 40 ml of three healthy study participants peripheral blood by venipuncture collected in sodium citrate vacutainer plastic tubes (BD)[21,31]. The monocyte fraction

was separated during density gradient centrifugation at 300 g at RT for 20 min, washed twice in phosphate-buffered saline solution (PBS, Lonza). Isolated cells were cultured on CELLCOAT Collagen Type 1-coated tissue culture flasks (Greiner Bio-One) using MCDB 131 medium (Gibco®, Life Technologies) containing proprietary supplements. An earlier cell passaging step seven days after initial isolation of the peripheral blood mononuclear cells was introduced to reduce expansion time and increase the final cell yield[67]. BOEC medium was changed every 2 days and cells were monitored till the appearance of the first colonies around day 14–28.

## Flow cytometry

BOECs were stained for 30 min at 4 °C in a dark place with Live Dead Pacific Blue dye (Life technologies) using 1 μL of dye every $1 \times 10^6$ cells resuspended in 1 mL of PBS. Live Dead dye is able to stain selectively dead cells in order to select population of living cells. Cells are stained for 20 min with primary antibodies in dark at 4 °C: anti-CD31-APC (1:100, Immunotools, 21270316), anti-CD34-PE (1:100, Immunotools, 21270344), anti-KDR-PE (VEGFR-2) (1:100, Miltenyi Biotech, 130-098-905), anti-TIE-2-APC (1:100, Miltenyi Biotech, 130-101-606), and anti-VE-Chaderin-PE (1:100, Miltenyi Biotech, 130-100-716). Samples are fixed with 200 μL cytofix/cytoperm buffer (ThermoFisher) for 20 min. Before every step, sample were washed with FACS Buffer and centrifugated at 1200 rpm for 8 min. Samples were acquired on a FACSCantoII instrument (BD Biosciences). Calibration beads (Life technologies) were used to calibrate and normalize acquisition settings in each experiment. Flow cytometry data were analyzed with FlowJo 10.6.2.

## Tube formation assay for BOEC functional assessment

The tube formation assay was performed according to manufacturer's protocols of Corning® Matrigel® Matrix. Briefly, Matrigel thawed overnight at 4 °C was mixed with VEGF (200 ng/ml) and 250 μL of matrix was added to each well of 24-well plates. After 1 h of incubation at 37 °C, cells ($10 \times 10^5$) were seeded onto the Matrigel. BOECs tube formation was observed for 16 h and performed in the presence of BOEC culture medium (BCM) based on MCDB 131 medium (Gibco®, Life Technologies) containing proprietary supplements, angiogenic medium[19] (AM), modified stabilization medium[19] (MSM) or the VEP two phase culture media protocol (11.5 h of angiogenic medium (AM), 70% of the culture time + 4.5 h of modified stabilization medium (MSM), 30% of the culture time respectively). To mimic the two-phase culture media protocol, all groups were exposed to the same medium firstly for 11.5 h and subsequently for 4.5 h. Cells were rinsed every media change. Analyses were performed at the end of the follow-up using an inverted phase-contrast microscope to investigate tube formation. ImageJ Angiogenesis software was used for the quantification of mesh size.

## Proliferation WST-1 assay

BOECs for WST-1 assay were seeded into 96 well plates at $1 \times 10^3$ cells/well. WST-1 assays were performed using Cell Proliferation Reagent WST-1 (Roche, Cat No. 11644807001). After 24 or 48 h of BOEC control medium (BCM) or VEP media protocol, the media was removed and cells were exposed for 30 min to the WST-1 reagent. Absorbance was measured at 450 nm using an ELISA plate reader.

## Isolation of neonatal pancreatic islets

Neonatal pancreatic islets (NPIs) were isolated from 1- to 7-day-old WT (66% males and 34% females) or CAG-iRFP (44% males and 56% females) transgenic piglets[22,68]. Briefly, pancreas was cut in 1 mm pieces, digested by collagenase-V (Sigma-Aldrich), sieved through a 500 μm mesh, and after 2 times wash, the isolates were cultured for 3 days in recovery medium (Ham's F12/M199 with protease inhibitors, antioxidants and additional nutrients). Full media change was carried out at day 1 and day 3 post isolation to remove degenerated exocrine cells. Subsequently, NPIs were maintained in maturation medium (Ham's F10, 10 mmol/L glucose, 50 μmol/L 3-isobutyl-1-methylxanthine, 0.5% [wt/vol] BSA, 2 mmol/L L-glutamine, 10 mmol/L nicotinamide, and 1% [vol/vol] penicillin/streptomycin stock, 1,6 mM $CaCl_2$) till the use in VEP or in insulin secretion test. Half of the medium was replaced every other day.

## Pig gene expression in NPIs and VEPs

To evaluate the endocrine maturation of NPIs in the VEP device, we performed gene expression experiments quantifying by ddPCR the pig *insulin* (Ss03386682_u1, ThermoFisher Scientific, Waltham, USA), *glucagon* (Ss03384069_u1, ThermoFisher Scientific, Waltham, USA), *somatostatin* (Ss03391856_m1, ThermoFisher Scientific, Waltham, USA) and *gapdh* (Ss03375629_u1, ThermoFisher Scientific, Waltham, USA) transcripts[8] at day 1 and day 7. The probe assays targeting the pig *insulin* and the human *GAPDH* (Hs04420697_g1, ThermoFisher Scientific, Waltham, USA) genes were also used to quantify the pig and human DNA, respectively, and to normalize the gene expression results based on the cell content of each VEP lysate. BOECs were used as an internal control of human DNA detection. VEPs at day 1 and day 7 were segmented in four subunits, lysed in 1 ml of Homogenization Solution from the Maxwell® RSC simplyRNA Cells Kit (Promega, Madison, USA) and then stored aliquoted at −80 °C. Each lysate was aliquoted into 200 μl aliquots ($n = 2$) for RNA extraction and 300 μl aliquots ($n = 2$) for DNA extraction. The Maxwell® RSC Instrument (Promega) was used to extract the RNAs with the Maxwell® RSC simplyRNA Cells Kit and the DNA with the Maxwell® RSC Blood DNA Kit (Promega). The RNA and DNA in each sample were quantified by the Quantus™ Fluorometer (Promega), using the QuantiFluor® RNA System or the QuantiFluor® ONE dsDNA System, respectively. A total of 25 ng of RNA were reverse transcribed using the High Capacity cDNA Reverse Transcription Kit (ThermoFisher Scientific), according to the random hexamers protocol. Different amounts of cDNA (0.08–4 ng) were tested for the indicated pig and human transcripts by ddPCR[44], and the measured copies were corrected by the RNA input in each reaction. Similarly, 20 ng of each DNA sample obtained from VEPs and BOECs were tested by ddPCR using the reported pig or human-specific DNA assays. The measured copies were corrected for the DNA input in the reaction, the total amount of DNA extracted from the sample and the starting extraction volume, obtaining an estimate number of cells present in each lysate. The copies per ng of RNA were normalized by cell content of each sample and differences over time were reported as fold change on day 1 expression.

## Dynamic insulin secretion test of NPIs

To assess insulin secretion after in vitro culture of NPIs, an insulin secretion test (IST) was performed on day 7 of culture[19]. Briefly, 100 NPIs were loaded in a perfusion chamber for suspension cells and connected to a high-capacity, automated perifusion system (BioRep® Perifusion V2.0.0). A low pulsatility peristaltic pump was used to push HEPES-buffered solution (125 mM NaCl, 5.9 mM KCl, 2.56 mM $CaCl_2$, 1 mM $MgCl_2$, 25 mM HEPES, 0.1% BSA, pH 7.4) through a sample of NPIs. The flow rate was set to 0.1 mL/min and the assay was performed at 37 °C environment. NPIs were initially perfused at 0 mM glucose for 40 min and subsequently with a low glucose insulin-free perifusion buffer (2 mM glucose) for 20 min to allow islet equilibration and removal of culture medium. Next, islets were stimulated with insulin-free perifusion buffer at 20 mM glucose for 20 min. Samples of the effluent were collected at baseline and at minutes 1 to 20 of stimulation. All samples were frozen at −80 °C until assayed for insulin content (Mercodia Insulin ELISA Kit, Mercodia). The dynamic insulin release results were compared using the following parameters: insulin release

as determined by the area under the curve (AUC, calculated by the linear trapezoid method expressed in fold changed over the basal of the insulin release), first insulin phase (mean of the peak at interval +2/+5 min), and second insulin phase (mean of the peak at interval +5/+8 min).

## In vitro seeding and culture of VEPs

Decellularized left lung lobes, from 6 to 8 weeks old male Lewis rats (175–200 gr), were mounted in a custom-made bioreactor with PA cannula and PV cannula attached to individual perfusion circuits and the trachea cannula attached to a reservoir that was about 25 cm (20 mmHg) above the level of the PA[19]. The trachea was kept open to the inside of the bioreactor and closed to the reservoir through a three-way valve 5 cm above the level of PA. For endothelial delivery through the PA and PV, 30 million BOECs were re-suspended in two separate seeding chambers (each with 20 million) in 80 ml of AM and seeded simultaneously through the PA and PV under 30 mmHg gravity. The organ was cultured under static condition for two hours. Perfusion was initiated at 0.5 mL/min from both the PA and PV. For tracheal seeding, the PV cannula was released and the trachea opened to the reservoir and closed to the chamber on day 1 of culture. 4000 NPIs were re-suspended with 10 million of BOEC in 50 ml of AM and seeded from the tracheal reservoir, followed by 200 ml AM. Static culture for 2 h completed airway seeding and before starting perfusion. To re-initiate perfusion, the trachea was opened to the chamber and flow was set to 1 mL/min from the PA. VEPs were cultured for a total of 7 days with the initial 5 days in AM and subsequent 2 days in MSM at 37 °C in 5% CO₂. As control for ex vivo and in vivo studies, VEP in the absence of BOEC (VEP⁻ᴮᴼᴱᶜ) and seeded with 4000 NPIs cultured for 7 or 21 days respectively, were generated following the same VEP seeding steps in the absence of BOECs. VEPs were harvested for functional and histological assessment on day 7 or 14. For in vivo transplantation VEPs were used at day 7.

## Fluorescence microangiography

Fluorescence microangiography was performed on matured VEPs in the absence of NPIs (VEPs⁻ ᴺᴾᴵˢ), which were submerged in PBS at 37 °C. 5 mL of 1% low melting point agarose (Invitrogen) in PBS containing 1:10 diluted blue FluoSpheres (0.2 µm, 365/415 Invitrogen) was manually injected into the PA (about 5 mL/min). PA perfusion pressure was below 20 mmHg (PressureMAT Single-Use Sensor). Immediately after microangiography, VEPs⁻ ᴺᴾᴵˢ were moved to ice, to allow the agarose to solidify[19,69].

## Insulin secretion tests (IST) of VEPs

VEPs and VEPs⁻ᴮᴼᴱᶜ cultured for 7 or 14 days received 4000 NPIs from the same batch. We tested for insulin secretion immature VEPs after 7 or 14 days of culture. VEPs were mounted on a custom-built ex-vivo perfusion device as followed: the VEP was placed on a platform in prone position with the PA cannula connected to a perfusion line, the PV cannula open to the level of the PA allowing collection of venous drainage, and with the trachea open 5 cm above the PA. Insulin secretion of VEPs was measured using a modified IST protocol[19]. VEPs were perfused with the perifusion buffer without glucose for 40 min to eliminate the insulin stored during the previous 7 days of culture, later at 2 mM glucose for 20 min to allow equilibration. Next, islets were stimulated with 20 mM glucose for 20 min. Samples were collected from venous drainage at baseline and at 1 to 20 min of stimulation. The IST was performed with 1 mL/min perfusion at 37 °C and 5% CO₂. All samples were frozen at −80 °C until assayed for insulin content (Insulin ELISA Kit, Mercodia). The dynamic insulin release results were compared using the following parameters: insulin release as determined by the area under the curve (AUC calculated by the linear trapezoid method expressed in picograms insulin/equivalent islet), first insulin phase (mean of the peak value at

interval 0/+5), and second insulin phase (mean of the peak value at interval +5/+8).

## MiR-375 quantification

MiRNA-375 (miR-375) is the most abundant miRNA in pancreatic islets, with an important role in maintaining the normal β cell mass[42,43]. Given the stability of miRNAs in the blood and the availability of highly sensitive technologies like the droplet digital PCR (ddPCR) for their measurement[44], miR-375 has been considered a good biomarker to track the β cell death in pre-diabetes and a predictor of endocrine dysfunction[45,46]. Here, we measured the miR-375 release and quantified the β cell loss during the VEP maturation. To rule out biases related to the miRNA contribution of BOECs, miR-375 was also quantified in a VEP seeded with BOECs only. Briefly, supernatants were sampled from 100 mL of VEPs culture media in DNA/RNA LoBind microcentrifuge tubes (Eppendorf®, Hamburg, Germany), stored at −80 °C and then extracted using the NucleoSpin miRNA Plasma kit (Macherey-Nagel, Düren, Germany), according to the manufacturer's instructions. 100 NPIs and 100,000 BOECs were pelleted and stored at −80 °C, lysed in ML lysis buffer from the Nucleospin miRNA kit (Macherey-Nagel, Düren, Germany), used for the subsequent small + large RNAs extraction. Eluted RNA from cells was quantified using the Epoch Microplate Spectrophotometer (BioTek Instruments, Inc., Winooski, VT, USA). cDNA was obtained by reverse transcription of 5 µL or 10 ng of extracted RNA from supernatants or cells miRNA respectively, using the TaqMan® MicroRNA Reverse Transcription kit (Applied Biosystems, Foster City, USA). A mix of diluted reverse-transcription primers, specific for human miR-375 (hsa-miR-375: assay ID 000564 Life Technologies, Carlsbad, USA) and miR-16 sequences (hsa-miR-16: assay ID 000391), was used as 5× reverse transcription primer, as indicated by the manufacturer's protocol. miR-375 and miR-16 (as quality control) were quantified by droplet digital PCR (ddPCR) using the QX100 ddPCR system (Bio-Rad, Hercules, USA) as already described[44]. The optimal amount of cDNA for each ddPCR was previously established and corresponded to 5 µl and 2 µl of cDNA from supernatants and to 0.5 ng and 1 ng of cDNA from cells for miR-375 and miR-16, respectively. The concentration of target miRNA copies in the supernatant was obtained by correcting for the starting volume and for the fraction of extracted RNA and cDNA used in each reaction. Target miRNA copies per cell were determined by correcting for the ng of RNA input, the total amount of extracted RNA and the number of cells in the original sample. Based on the estimated number of miR-375 copies per cell and the miR-375 copies measured in the VEP supernatants at each time-point, we calculated the percentage of β cells lost during the VEP maturation process.

## Diabetes induction and metabolic monitoring

Diabetes was induced by administrations of alloxan (72 mg/kg i.p., Sigma-Aldrich) in 6–8 weeks female NSG mice (22–25gr) on day −2 before transplantations[19,70–72]. Animals with non-fasting glucose <450 mg/dl prior transplantation were not considered diabetic and excluded from the study. The reversal of diabetes was defined as two consecutive glycemia measurements ≤200 mg/dL and maintained until study completion. Islet graft function was assessed daily post islet transplantation through non-fasting blood glucose measurements, using a portable glucometer (OneTouch Ultra, Bayer). Mice died for surgical complications within first week were not considered for Kaplan Maier analysis. Exogenous insulin therapy (LinBit pellet; Lin-Shin) was administered subcutaneously peri-transplant to maintain an acceptable health status[52]. The day after transplantation a suboptimal quantity of two pellets were implanted in all diabetic recipients with a not fasting glycemia >250 mg/dL. No pellets were implanted in animals with a sustained function and not fasting glycemia ≤250 mg/dL. One pellet was removed at ten days after transplantation and the second one twenty days after transplantation.

### Device-less (DL), Kidney capsule (KC), and Liver (LV) NPIs transplantation

NSG 6–8 weeks female NSG mice (22–25gr) were anesthetized with Avertin (tribromoethanol) (intraperitoneal injection at a dose 0.75 mg/g; Sigma). As described[52], 3 to 6 weeks before islet transplantation, the device-less (DL) site was created by implanting a 2-cm segment of 5-French (Fr.) textured nylon radiopaque angiographic catheter (Cook Medical, Indiana, USA) subcutaneously into the lower left quadrant of recipient NSG mice. At the time of transplantation, a small (4 mm) incision was made to gain access to the catheter, and the tissue matrix surrounding the superior margin of the catheter was cut to remove the catheter. A total of 4000 NPIs were placed into polyethylene (PE-50) tubing and infused within DL space using a microsyringe (Hamilton). The incision was closed with a surgical silk suture (Ethicon, New Jersey, USA). In addition, a subset of diabetic animals was transplanted with 4000 NPIs/recipient under the KC or into LV via the portal vein, the standard sites for rodent islet transplantation[73,74]. Due to surgical limitation, LV accepted only 800 NPIs for each NSG diabetic recipient. For all experiments, NPIs were pooled, batched, and transplanted to either the DL, KC, or LV sites. DL and LV transplanted mice were followed till 9 weeks whereas KC till 14 or 18 weeks after implantation.

### VEPs implantations in diabetic recipient mice

A total of 6–8 weeks female NSG mice (22–25gr) were anesthetized with Avertin. Prior to mice implantation, VEPs and VEPs$^{-BOEC}$ were flushed with 250 mL ice-cold DMEM (GIBCO- 5.5 mM glucose) from the pulmonary artery (PA). Next, VEPs were carefully dissected from the heart and trachea and divided with sagittal incision in two segments. VEPs segments were implanted in two separate subcutaneous pockets generated on the dorsal left and right area. The single pocket was specifically created by two 10-mm lateral transverse incisions limiting a 2 cm tunneled area. An adequate void (1 cm by 2 cm) was created. Once VEPs were placed in the dedicated space, the implant was ligated under the skin with two prolene sutures (Ethicon, New Jersey, USA) by sides and skin incision was closed with surgical silk suture (Ethicon, New Jersey, USA). Transplanted mice were followed up till 9, 14, or 18 weeks after implantation.

### Oral glucose tolerance test (OGTT)

An oral glucose tolerance test (OGTT) was performed on NSG transplanted with VEPs, DL-NPIs, KC-NPIs, and LV-NPIs 9 weeks after transplantation: 1 g/kg of glucose was administrated by oral gavage after 4-h fast[75]. Blood glucose was determined at 0, 10, 20 30, 60, 90, and 120 min after glucose administration. The area under the curve (AUC) of glucose during OGTT was calculated using the trapezoidal method (baseline = 0 min).

### VEPs vascular density analysis by AngioTool

VEPs segments at 1, 2, 4, 9, 14, and 18 weeks after implantation were explanted and 32-bit color images were acquired. Explant samples were excluded from the analysis in the presence of surgical damage. The images were processed by using the unbiased AngioTool software[27], generating an analysis that includes segmentation, skeletonization and analysis of the vasculature. On VEPs image, AngioTool 64 version 0.6a identifies vessel profiles according to the software's preset parameters. Identified vessels are demarcated with an outline on the displayed image which dynamically updates its shape in response to adjustments performed using the controls included in the analysis tab. Once the outline overlay matches the vessels in the displayed image, the analysis can be carried out. On completion of the analysis, the resulting image shows an overlay, which indicates the area encompassing all vessels, a skeletal representation of the vascular network and the computed branching points inside this area. This image that represents the explant area, the outline of the analysis, the vessels and their branching point, is saved together with an Excel file containing the analysis parameters and the computed results (explant area, vessel area, percentage surface vessel density, total number of junctions, junction density, total vessel length, average vessel length).

### Detection of CAG-iRFP transgenic NPIs by fluorescence imaging

Detection of fluorescent images (FI) of CAG-iRFP transgenic NPIs were performed using an IVIS® Spectrum bioluminescent in vivo imaging system (PerkinElmer), calibrated to enable absolute quantitation of the bioluminescent signal and longitudinal studies performed over many time points[76]. Transplanted iRFP-NPIs in DL, KC, LV and VEPs were acquired by placing the NSG recipient mice at 37 °C under gaseous anesthesia (2–3% isoflurane and 1 l/min oxygen). In vivo FIs were performed by IVIS® SpectrumCT System (Perkin Elmer), equipped with a low noise, back-thinned, back-illuminated CCD camera cooled at −90 °C and a with quantum efficiency in the visible range > 85%. Images were obtained using the following settings: exposure time=auto, binning=8, f = 2 and a field of view equal to 13 cm (field C); when needed spectral unmixing, was obtained using the following excitation/emission filters: 640/680, 640/700, 640/720, 640/740, 640/760, 675/720, 675/740, 675/760, 675/780, 675/800 nm. To track the persistency of iRFP signal, graft FIs were acquired prior implantation and at the end of the follow up. Image analyses were carried out considering two similar region of interest (ROI) one placed over the kidney (Tiss) and a background (Bk) region near to the implantation site. The radiant efficiency within this ROI's was measured using images acquired with the 675/720 filters. The tissue to background ratio TB (ti) was then calculated at different time points (ti) as follow: TB(ti) = [Tiss(ti)]/(Bk(ti)]. Spectral un-mixing of the FI data was performed on selected time points in order to show the specificity of the fluorescence signal over the tissue auto-fluorescence. All the images were acquired and analyzed using Living Image 4.5 (Perkin Elmer).

### Histology and immunofluorescence staining

All samples were fixed overnight with 4% paraformaldehyde in PBS (PFA) unless stated otherwise. For paraffin-embedded samples, 5 μm sections were used for histological analysis. For immunofluorescence staining, sections were deparaffinized, rehydrated, put through heat-induced antigen retrieval, permeabilized with 0.1% Triton X-100 in PBS, washed three times in PBS and blocked with 1% bovine serum albumin in PBS. Sections were incubated in primary antibodies at 4 °C. Primary antibodies included antibodies against human CD31 (1:40, DAKO, M0823, clone: JC70A), human vWF (1:200, DAKO, A0082, polyclonal), human VE-Cadherin (1:40, R&D system,AF938, polyclonal), human VEGF-R2 (1:200, Cell Signaling, 2479 S, polyclonal), pig Insulin (1:200; DAKO, A0564, polyclonal), pig Chromogranin A (1:200, Abcam, AB15160, polyclonal), mouse CD31(1:300, Biolegend, 102501, clone: MEC13.3), mouse CD11b-PE (1:100, Biolegend, 101208, clone: M1/70), mouse LY6G-AF647(1:100, Biolegend, 127610, clone: 1A8), mouse anti-tubulin β3-AF647 (1:100 Biolegend, 801210, clone: TUJ1), anti-pig somatostatin (H-11) (1:100 Santa Cruz Biotechnology, Sc-74556, clone H-11), anti-pig glucagon (1:200 Invitrogen, PA5-83353, polyclonal). A546-, A488- or A647-labelled secondary antibodies and DAPI (1:500; Molecular Probes) for 40 min at room temperature, followed by three washes in PBS. Sections were mounted with Mount Quick (Bio-optica) and imaged on a GE healthcare DeltaVision™ Ultra microscope and Olympus FluoVIEW FV3000RS Confocal. Microangiography samples were fixed in PFA for 2 h, equilibrated in 30% sucrose in PBS overnight, and embedded in O.C.T. compound (Tissue-Tek). A total of 25 μm cryo-sections were prepared for staining. Sections were mounted with Olympus FluoVIEW FV3000RS Confocal. Insulin, glucagon and somatostatin fluorescence intensity at day 1 and day 7 were analyzed from NPIs or in VEP slices. The degree fluorescence intensity was analyzed

by two independent observers who were blinded to the experimental conditions. ImageJ version:2.1.0/1.53C software package was used for image analysis and fluorescence quantifications[77].

## Transmission electron microscopy (TEM)

For transmission electron microscopy, 14-week VEP grafts were fixed 24 h at 4 °C with 4% paraformaldehyde and 2.5% glutaraldehyde in 125 mM cacodylate buffer. Subsequently, it was washed in cacodylate buffer for four times followed by post-fixation with 1% osmium tetroxide in 0.1 M cacodylate buffer for 2 h at 4 °C. Samples were dehydrated for 30 min and embedded in Epon 812. Then, ultra-thin sections were cut at 70 nm and contrasted with uranyl acetate and lead citrate and examined with a ThermoFisher Talos L120C electron microscope.

## Statistics and reproducibility

Values are summarized as mean ± SD (or SEM, where indicated) or median according to their distribution. Statistical analysis was performed by Mann Whitney U tests, Student's t-tests (two-tail comparisons), ANOVA (with Post-hoc correction) and general linear model. Gain of normoglycemia was evaluated by Kaplan-Meier analysis and the significance was estimated using the log-rank test. Statistically significant differences were defined as $p < 0.05$ and $p$ value is presented in each graph. The $n$ number for each graph is provided in the corresponding figure legend. Microsoft Excel 2013 version 15.0.5501.1000 (Microsoft Corporation, https://www.microsoft.com), Graphpad Prism v9 version 9.4.1 (GraphPad Software, http://www.graphpad.com) and R version 4.2.2 (R Core Team (2020) R: A Language and Environment for Statistical Computing. R Foundation for Statistical Computing, Vienna, Austria; rstatix and survminer packages) were used for data management, statistical analysis and graphs generation.

## Reporting summary

Further information on research design is available in the Nature Portfolio Reporting Summary linked to this article.

## Data availability

All data generated or analysed during this study are included in this published article (and its supplementary information files). Source data are provided with this paper." Source data are provided with this paper.

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

## Acknowledgements

This study was supported by a grant from Juvenile Diabetes Research Foundation (JDRF 1-SRA-2019-771-S-B) (L.P. and A.C.), "I love riccio" fundraising campaign, SOStegno 70 Insieme ai ragazzi diabetici Associazione Onlus (Project "Beta is better") (L.P.). In addition, the study was supported by the Deutsche Forschungsgemeinschaft (TRR127; E.W., E.K.), by the European Union's Horizon 2020 research and innovation program under grant agreement No. 760986 (iNanoBIT) (E.W., E.K.), and by the German Federal Ministry of Education and Research (BMBF) to the German Centre for Diabetes Research (DZD e.V.) (Grant No. 82DZD00802, E.W., E.K.). L.V. received funding from the European Union's Horizon 2020 research and innovation program under the Marie Skłodowska-Curie grant agreement No. 812660 (DohART-NET). The authors thank Paola Macchieraldo, Antonio Mincione, Elena Riva, Antonio Civita, Andrea Marchesi and Michele Mainardi for supporting the fundraising campaign "Un brutto t1po". The authors thanks Prof. Carlo Tacchetti, Director of the Experimental Imaging Center, for the support in TEM evaluation and interpretations. Immunofluorescence and TEM acquisition were carried out in ALEMBIC by San Raffaele Scientific Institute. The authors thank Rachel C. Applefield for her help in the English language revision of this paper.

## Author contributions

A.C. design, conducted, and analyzed all experiments, and wrote the manuscript; A.N., C.P., and F.C. assisted with organ in vitro preparation and analysis. M.M., M.P., S.P., and E.D. assisted with in vivo experiments and contributed to manuscript preparation. M.C.M. assisted with preliminary experiments. F.M. assisted with immunofluorescence acquisition. I.M. and V.L. performed miRNA analysis and data interpretations. C.O., A.C., and A.F. performed BOEC isolation, characterization, validation and in vitro testing. L.V., E.K., and E.W. contributed to NPIs isolation and characterization. L.P. oversaw the study and revised manuscript.

## Competing interests

The authors declare no competing interests.
