## [Peer Review File · Nature Communications]

Directed Self-Assembly of a Xenogeneic Vascularized Endocrine Pancreas for Type 1 DiabetesREVIEWER COMMENTS

Reviewer #1 (Remarks to the Author):

In the manuscript "Directed self-assembly of a xenogeneic vascularized endocrine pancreas for type 1 diabetes" Citro et al have previously proposed the use of decellularized lung as scaffold for the beta cell replacement therapy with the final goal to engineer a functional vascularized endocrine organ. In the present study the authors have integrated neonatal porcine islet cells and patient specific blood outgrowth endothelial cells to engineer a xenogenic-vascularized endocrine pancreas. They have validated ex vivo VEP cell integration and function, in vivo VEP engraftment and performance in a preclinical model of diabetes. Their results show that VEP enables functional endocrine maturation of neonatal porcine islet cells in vitro and in vivo function subsequent to transplantation. One interesting aspect pointed out by the authors is that given recent progression in genetic engineering of neonatal porcine islet cells donor pigs, the actual technology may enable assembly of immune-protected functional personalized endocrine organs.

This is an interesting manuscript where especially the transplantations and evaluation of in vivo function hold a lot of promise. There are however a few things that should be considered before publication. There should be a more detailed in vitro description of the matured neonatal porcine islet cells in terms of immuno-histochemistry and function. With regard to the in vivo situation how do the capillaries look like at the EM level. Are they fenestrated? Are the neonatal porcine islet cells becoming innervated in vivo. The authors state that the median time for gaining normoglycemia was two days after VEP transplantation. How can this be explained in terms of vascularization and innervation. What happens with VEP over time, i.e. beyond the 60 days. Will VEP still be functional? Also, what happens with surface vessel density over time?

Reviewer #2 (Remarks to the Author):

This paper describes the use of a decellularized lung (rat) with blood outgrowth endothelial cells (BOEC, human) and neonatal pancreatic islets (NPI, pig) to create a vascularized subcutaneous insulin secreting device (termed VEP). Ignoring the multi-species sourcing problems, once I figured out what the acronyms meant I found the data and story quite compelling. Subcutaneous vascularization using EC for the purpose of islet transplantation is still novel although there are now many reports of its utility. Using a decellularized lung as the scaffold for this purpose is interesting and shows compelling results, in terms of both insulin kinetics and NPI transplant outcomes. NPI need to mature in situ and so there is generally a delay in onset of normoglycemia, although this was not seen here. I am not clear I understood why – was this because of BOEC? In the discussion, it would be helpful to highlight the issues with this approach and the challenges that the applicants foresee in moving towards the clinic – questions of immune response for example along with the challenge of manufacturing such a device.

I am in favour of accepting this paper for publication. Although I have a few minor comments.

1. It was not totally clear until I read closely that the BOEC were placed the blood compartment of the lung, while the NPI (and more BOEC) were on the alveoli side. This becomes obvious from eventually Fig

- 2, but a few words in the introduction would be helpful, as would more labels on Fig 2. A schematic of the “final device” would also be helpful
2. The article was a little hard to follow in places due to the use of acronyms and the absence of paragraph breaks in some sections such as the discussion (called conclusion in error). It would also be helpful to make it clear that the same number of NPI (4000) were used in all cases.
3. Recellularization of decellularized organs is thought to be a challenge. Can the authors comment on how many BOEC/NPI (or yield relative to what was delivered) were actually deposited/embedded
4. A glycemia threshold of 250 mg/dL is a bit high, although not uncommon in the literature. Would the results look different if a tighter criterion was used. Would fig 3C be less clear?
5. Image analysis for the iRFP NPI was unclear. What was the kidney tissue for – only for the kidney capsule. What was done for the subcu implants?
6. Fig 2 – all refer to the same number of NPI?
7. Explain in methods and results why miR-375 was measured. This was unclear and results in Figure 2 (eg panel G) were not well described. Panel G shows that NPI are dying? And how does this match up to panels H and I
8. VEP had immediate graft function – this was attributed to maturation of the BOEC. Was this a typo and should have been maturation of the NPI?
9. I am not a big fan of the % vascular surface metric. Can these numbers be converted into an average vessel density or an average distance of vessel to cell.
10. The “conclusion” refers to a VEP implant without BOEC but data is not shown. These results would be most interesting and a useful control for the addition of BOEC.
11. What does “also exocrine cells” referring to NPI mean?
12. To what extent is aVEP based on a decellularized lung, a “personalized device”

Reviewer #3 (Remarks to the Author):

No comments.

A Reviewer #1 (Remarks to the Author)

In the manuscript "Directed self-assembly of a xenogeneic vascularized endocrine pancreas for type 1 diabetes" Citro et al have previously proposed the use of decellularized lung as scaffold for the beta cell replacement therapy with the final goal to engineer a functional vascularized endocrine organ. In the present study the authors have integrated neonatal porcine islet cells and patient specific blood outgrowth endothelial cells to engineer a xenogenic-vascularized endocrine pancreas. They have validated ex vivo VEP cell integration and function, in vivo VEP engraftment and performance in a preclinical model of diabetes. Their results show that VEP enables functional endocrine maturation of neonatal porcine islet cells in vitro and in vivo function subsequent to transplantation. One interesting aspect pointed out by the authors is that given recent progression in genetic engineering of neonatal porcine islet cells donor pigs, the actual technology may enable assembly of immune-protected functional personalized endocrine organs.

This is an interesting manuscript where especially the transplantations and evaluation of in vivo function hold a lot of promise.

We thank the Reviewer for the summary of our work and her/his kind comment.

There are however a few things that should be considered before publication. There should be a more detailed in vitro description of the matured neonatal porcine islet cells in terms of immunohistochemistry and function.

In agreement with the Reviewer's comment, a more detailed description of the matured neonatal porcine islet cells in terms of immunohistochemistry and function is now reported in the revised manuscript. Insulin, glucagon, and somatostatin were evaluated during VEP maturation by immunofluorescence and ddPCR. Batch matched NPIs in standard culture or in VEP were analyzed at day 1 and 7 and the data are now reported in Figure 2 (panel D-G) and Supplementary figure 2. Higher levels of insulin were shown as both protein and mRNA in NPIs in VEP at 7 days compared to VEP at day 1. Additionally, higher fluorescence levels of insulin and glucagon were evident in NPIs seeded in VEP compared to batch matched NPIs in standard cultures at day 1 and 7 (Fig. 2E-G), confirming that the presence of the vascularized ECM is an inducer of the maturation. In fact, the insulin secretion of NPIs matured in VEP was higher than that of NPIs kept in standard culture (Fig. 3A-B). Moreover, the extension of the maturation of NPIs in VEPs up to 14 days resulted in the further improvement of the biphasic insulin secretion (Fig. 3C). The presence of endothelial cells appears relevant in this maturation process, since the seeding of NPIs in the VEP in the absence of BOEC determines a lower insulin secretion (supplementary Fig. 3).

With regard to the in vivo situation how do the capillaries look like at the EM level. Are they fenestrated? Are the neonatal porcine islet cells becoming innervated in vivo?

To address the point raised by the reviewer, we performed the transmission electron microscopy (TEM) analysis of VEP. We consider 14 weeks an adequate time to evaluate the VEP vascular and nerve status. Vessels (not fenestrated) filled with red blood cells and nerves in close contact with granulated endocrine cells were evident. These data are now reported in supplementary figure 7

The authors state that the median time for gaining normoglycemia was two days after VEP transplantation. How can this be explained in terms of vascularization and innervation.

We thank the Reviewer for this comment. The early function is the results of two different factors. As previously reported (PMID: 30735895) the use of vascularized lung scaffold is able to improve the islet engraftment thanks to the early connection between the donor and recipient vascular network. In VEP, a second factor is the direct role of BOEC on NPI maturation. As reported before, the seeding of NPIs in the VEP in the absence of BOEC determines a delay in maturation associated with a lower insulin secretion capacity (supplementary Fig. 3). To try to understand the relative impact of these two factors, we generated VEPs using NPIs previously fully matured in vitro by culturing for 21 days to achieve the full insulin secretion capacity (PMID: 29975241). In the absence of BOEC, once implanted in vivo, we still observed a four week delay in the time of engraftment (supplementary Fig. 5). This result confirms that the presence of endothelial cells represents a key factor in enabling the VEP's immediate engraftment and function. As discussed before, we reported that implanted VEP showed nerves, both in immunofluorescence and TEM, in the graft surface edges and in the proximity of granulated endocrine cells at 14 weeks (supplementary figure 7). Despite this, no innervation was observed 1 week after implantation (data not shown) and we can exclude a role of innervation in early VEP function.

What happens with VEP over time, i.e. beyond the 60 days. Will VEP still be functional? Also, what happens with surface vessel density over time?

The main goal of our in vivo work was to study the engraftment and the short term function of VEP and for this reason we have planned a follow up of 8-9 weeks (~60 days). As requested by the reviewer, we performed new experiments to study the long term function of VEP. The follow up was extended to 18 weeks, an adequate timeframe for evaluating the long-term function according to the recent publications in the field (PMID: 31601796; PMID: 34078744; PMID: 36229614; PMID: 35288694; PMID: 33316848). More in detail, we extended the follow up of VEP and NPI in kidney capsule, as internal control, until 14 or 18 weeks. The mice implanted with VEP achieved normoglycemia within the first three days and maintained it throughout the entire follow up period (Fig. 4B-C). To track insulin secreting cells using live imaging, some mice received iRFP NPIs within VEP. Consistently with the achievement and maintenance of normoglycemia, the presence of positive IVIS signal confirmed the graft persistency until the end of the follow-up (Fig 5A-B). The VEP explants at 9, 14 and 18 weeks were always associated with the occurrence of hyperglycemia (Fig.4B), confirming that normoglycemia was VEP dependent. VEPs were analyzed in terms of vessels density. We observed a trend in increasing the vessels density from 9 to 14 or 18 weeks, even if the most significant increase develops within the first 14 weeks (Fig. 5E). In agreement with the literature, NPIs in kidney capsule started to restore normoglycemia from 6 to 9 weeks after implantation and preserve the established function until the end of the follow up (from 9 to 14 or 18 weeks) where we found a comparable function to the VEP group (Fig. 4B-C).

Reviewer #2 (Remarks to the Author)

This paper describes the use of a decellularized lung (rat) with blood outgrowth endothelial cells (BOEC, human) and neonatal pancreatic islets (NPI, pig) to create a vascularized subcutaneous insulin secreting device (termed VEP). Ignoring the multi-species sourcing problems, once I figured out what the acronyms meant I found the data and story quite compelling. Subcutaneous

vascularization using EC for the purpose of islet transplantation is still novel although there are now many reports of its utility. Using a decellularized lung as the scaffold for this purpose is interesting and shows compelling results, in terms of both insulin kinetics and NPI transplant outcomes.

We thank the Reviewer for the summary of our work and her/his kind comment.

NPI need to mature in situ and so there is generally a delay in onset of normoglycemia, although this was not seen here. I am not clear I understood why – was this because of BOEC?

We thank the reviewer for this comment and for giving us the opportunity to clarify this point. In our work, we demonstrated that VEP ex vivo maturation in the bioreactor system is a strong boost for NPIs functional maturation. In agreement with the Reviewer 1 comment, more data are now included in the revised manuscript. Insulin, glucagon, and somatostatin were evaluated during VEP maturation by immunofluorescence and ddPCR. Batch matched NPIs in standard culture or in VEP were analyzed at day 1 and 7 and the data are now reported in Figure 2 (panel D-G) and Supplementary figure 2. Higher levels of insulin were shown as both protein and mRNA in NPIs in VEP at 7 days compared to day 1. Additionally, higher levels of insulin and glucagon were evident in NPIs seeded in VEP compared to batch matched NPIs in standard cultures at day 1 and 7 (Fig. 2E-G), confirming that the presence of the vascularized ECM is an inducer of the maturation. The insulin secretion agrees with the expression data. In fact, the insulin secretion of NPIs matured in VEP was higher than that of NPIs kept in standard culture (Fig. 3A-B). Moreover, the extension of the maturation of NPIs in VEPs up to 14 days resulted in a further improvement of the biphasic insulin secretion (Fig. 3C). The presence of endothelial cells appears relevant in this maturation process, since the seeding of NPIs in the VEP in the absence of BOEC determines a lower insulin secretion (supplementary Fig. 3). A second factor is relevant for early in vivo function. As previously reported (PMID: 30735895) the use of vascularized lung scaffold is able to improve the islet engraftment thanks to the early connection between the donor and recipient vascular network. To try to understand the relative impact of these two factors, we generated VEPs using NPIs previously fully matured in vitro by culturing for 21 days to achieve the full insulin secretion capacity (PMID: 29975241). In the absence of BOEC, once implanted in vivo, we still observed a four week delay in the time of engraftment (supplementary Fig. 5). This result confirms that the presence of endothelial cells represents a key factor in enabling the VEP's immediate engraftment and function.

In the discussion, it would be helpful to highlight the issues with this approach and the challenges that the applicants foresee in moving towards the clinic – questions of immune response for example along with the challenge of manufacturing such a device.

As requested by the reviewer a dedicated comment on moving towards the clinic and immune response was included in the discussion

I am in favour of accepting this paper for publication.

We thank the reviewer in appreciating our work.

Although I have a few minor comments.

1. It was not totally clear until I read closely that the BOEC were placed the blood compartment of the lung, while the NPI (and more BOEC) were on the alveoli side. This becomes obvious from eventually Fig 2, but a few words in the introduction would be helpful, as would more labels on Fig 2. A schematic of the “final device” would also be helpful.

We thank the reviewer for suggesting this improvement in the clarity of the paper. Following the comment raised by the reviewer, we integrated a dedicated sentence in the introduction, more labels (i.e. NPIs and BOEC cells amount, legend for the media AM/MSM and decellularized rat lung access), and a new schematic of the “immature VEP” at day 0 and the “final VEP” structure in Figure 2A.

2. The article was a little hard to follow in places due to the use of acronyms and the absence of paragraph breaks in some sections such as the discussion (called conclusion in error). It would also be helpful to make it clear that the same number of NPI (4000) were used in all cases.

To address the reviewer’s comment, we introduced a paragraph dedicated to acronyms, paragraph breaks in some sections and we changed the paragraph called “conclusion” to “discussion”. Additionally, we introduce the number of NPIs used in all in vivo cases in Fig. 4A

3. Recellularization of decellularized organs is thought to be a challenge. Can the authors comment on how many BOEC/NPI (or yield relative to what was delivered) were actually deposited/embedded

*We agree with the reviewer about the challenge related to the organ re-cellularization process. To provide full endothelial coverage, once the BOEC behavior had been certified with VEP media protocol, we consistently adopted the same amount of endothelial cells (30 million through Pulmonary Artery and Pulmonary Vein + 10 million within Trachea) that allows the full decellularized lung vascular coverage that we published in 2019 (PMID: **30735895**). Additionally, 4000 absolute NPI clusters were seeded through the trachea in all VEP conditions. During the seeding process, at day 0 and day 1, no cell spillage from endothelial and endocrine compartments were observed.*

4. A glycemia threshold of 250 mg/dL is a bit high, although not uncommon in the literature. Would the results look different if a tighter criterion was used. Would fig 3C be less clear?

*We thank the reviewer for this comment. As confirmed by the reviewer, 250 or 200 mg/dl of glycemia threshold are commonly used in several preclinical models of islet transplantation (PMID: **31267721**; PMID: **28902069**; PMID: **23751893**). To rule out bias related to the threshold, we analyzed the data, and we reported the Kaplan Mayer analysis with a glycaemia threshold of 200 mg/dl (figure 4C). Also in this case, the updated analysis confirmed the significant differences between the 4 groups, supporting the role of VEP in reverting the hyperglycemic status. In agreement with the reviewer’s point of view, we decided to exchange Fig. 3C (now 4C) with the new version requested. Data were updated according to the reviewer 1’s request on the new follow up (18 weeks).*

5. Image analysis for the iRFP NPI was unclear. What was the kidney tissue for – only for the kidney capsule. What was done for the subcu implants?

We apologize if the iRFP NPIs figure was not clear. To solve this issue, we rename the groups KC-NPIs and LV-NPIs in Kidney Capsule-NPIs and Liver-NPIs to specify the acquisition of the NPIs transplanted in the Kidney Capsule and in the Liver. The IVIS analysis proposed in Fig.5A-B were performed in recipient NSG mice at 9, 14 or 18 weeks after implantation. To clarify this detail, we reported the

time point in the figure and within the legend. Based on the results obtained in our recent publication (PMID:34935207), in long term follow up only normoglycemic mice can showed iRFP signal at IVIS acquisition. In this scenario, as shown by the glycemia curve, no mice in the subcutaneous implants were normoglycemic at longer time points and no signal was detected. We decided to exclude this acquisition from the main figure to not crowd the final picture. We integrated 1 week and 9 weeks subcutaneous NPIs iRFP detection in supplementary Fig. 4. As shown by the IVIS detection, we tracked the iRFP signal in the subcutaneous implantation at an early time point (1 week) but, in agreement with the glycaemia profile, no signal was detected at 9 weeks.

6. Fig 2 – all refer to the same number of NPI?

We thank the reviewer for his/her request. Yes, in Fig. 2 (now Fig. 3) we cultured the same NPIs dose in vitro for VEP and standard condition. We specify the number of NPIs used in standard condition and in VEP fashion in the material and methods section.

7. Explain in methods and results why miR-375 was measured. This was unclear and results in Figure 2 (eg panel G) were not well described. Panel G shows that NPI are dying? And how does this match up to panels H and I

We apologize if the use of miR-375 was not clear. We introduced additional descriptions of the miR-375 quantification both in the methods and in the results sections. As mentioned in the manuscript, MiR-375 is a highly expressed microRNA in pancreatic β cells and previous studies suggested that it is a suitable biomarker for real-time detection of beta cell death. Since we wanted to track the preservation of the beta cell mass, we took advantage of miR-375 to monitor the beta cell death during the VEP maturation process. In Fig. 3D, we quantified the kinetics of miR-375 released by VEP as absolute number of copies. In Fig. 3E, based on the estimated number of miR-375 copies per cell we estimated the percentage of beta cells death during the VEP maturation process. The results suggested an estimated range of β cell death between 0 to 5% during the 7-days VEP maturation process

8. VEP had immediate graft function – this was attributed to maturation of the BOEC. Was this a typo and should have been maturation of the NPI?

We apologized for the mistake. As understood by the reviewer, we attribute the immediate graft function to the NPI functional maturation. We have corrected this error in the text.

9. I am not a big fan of the % vascular surface metric. Can these numbers be converted into an average vessel density or an average distance of vessel to cell.

While we understand the criticism raised by the reviewer, we kindly disagree that the data should be represented as average vessel density or an average distance of vessel to cell. In our case, in agreement with other publication (PMID: 33215738; PMID: 30310137), we think that the percentage of vessel density is informative of the progression of VEP vascular integration; on the other end, we report below, the graphs of Average Vessel length (mm) and Vessel length density (mm/mm²) that indicate a progressive improvement of the VEPs vessel structure over the time. If the reviewer considers it helpful, we can add these figures as supplementary information.

10. The “conclusion” refers to a VEP implant without BOEC but data is not shown. These results would be most interesting and a useful control for the addition of BOEC.

We thank the reviewer for the comment. The original sentence referred to the in vitro VEP function in the absence of BOEC (VEP^{BOEC}). Based on this suggestion, we add the preliminary analysis of VEP^{BOEC}, both ex vivo and in vivo, as supplementary information. The data shows that VEP^{BOEC} has a substantially reduced performance, compared to the fully matured VEP, both in terms of insulin secretion ex vivo (supplementary fig 3) and time of engraftment in vivo (supplementary fig. 6B) .

11. What does “also exocrine cells” referring to NPI mean?

As reported from literature, since NPI isolation is not performed in the presence of a gradient, acinar duct and progenitor cells are still present in the final product. These “contaminants” are progressively lost during the culture, but it is not possible to exclude the persistency of these components in the preparations. We agree with the reviewer that this can be misleading, and we removed the sentence from the discussion.

12. To what extent is a VEP based on a decellularized lung, a “personalized device”

We understand the point of the reviewer, and we changed the term “personalized device” to “flexible device,” related to the possibility to match, in different combinations, relevant criteria for the beta cell replacement field.

Reviewer #3

In this manuscript, Citro et al used decellularized rat lungs as scaffolds for neonatal porcine islets (NPI) mixed with human blood outgrowth endothelial cells (BOECs) to engineer an endocrine pancreas for beta cell replacement for T1D. Such engineered endocrine pancreas (VEP) with immature neonatal islets, when transplanted to diabetic female NSG mice, were able to provide immediate glycemic control, comparable to that with mature adult pig islets. The current study is based on the group's previous experience of engineering VEP, and now advancing to using NPIs as a step forward to clinical translation. Overall, the study was coherently designed and executed. The results have high translational value.

We thank the Reviewer for the summary of our work and her/his kind comment.

1. Were wild type pig NPIs used or NPIs of the several versions of genetically modified pigs (please clarify as such under the appropriate subsection in "Material and Methods")? Can they comment on possible differences in terms of maturation requirements and culture specifics using NPIs from various pig sources?

We thank the reviewer for this comment. We exclusively used NPIs isolated from WT or CAG-iRFP transgenic piglets. This information is reported in "Material and Methods" in "Animals" and now also in "Isolation of neonatal pancreatic islets" specific sections. We agree with the reviewer on the possible differences in terms of maturation requirements and culture specifics using NPIs from various pig sources and we added this comment in the discussion section.

2. It is not clear from the text or the figure/legends how many donors were experimented for BOEC isolation. Do they anticipate any variabilities from donor to donor? If so, how would such variabilities affect subsequent VEP engineering process?

We thank the reviewer for his/her comment. We now reported in the "Material and Methods" section that we isolate BOEC from three healthy donors. BOEC isolation and expansion was already extensively characterized (PMID: 34853801; PMID: 26780290; PMID: 19891540). As reported from the literature, we observed variability in the number of colony formation capacity but not in their performance during in vitro expansion and VEP engineering process. To reinforce this concept we added this comment in the discussion.

3. What is the explanation for an accelerated maturation of NPIs in the engineered VEP? Or, was the faster glycemic control simply a result of less NPI death (because of better vascularization?) in comparison to other transplant routes? In vitro characterization of NPIs on their maturation process during VEP engineering would be informative. However, it is also possible that an in vivo signal (or signals) is/are required for accelerated maturation. Effort to explain/investigate this unique observation should be attempted.

We thank the reviewer for this comment and for giving us the opportunity to clarify this point. In our work, we demonstrated that VEP ex vivo maturation in the bioreactor system is a strong boost for NPIs functional maturation. In agreement with the Reviewer 1 and 2 comment, a more detailed in vitro description of the matured neonatal porcine islet cells in terms of immuno-histochemistry and function is now reported in the revised manuscript. Insulin, glucagon, and somatostatin were evaluated during VEP maturation by immunofluorescence and ddPCR. Batch matched NPIs in

standard culture or in VEP were analyzed at day 1 and 7 and the data are now reported in Figure 2 (panel D-G) and Supplementary figure 2. Higher levels of insulin were shown as both protein and mRNA in NPIs in VEP at 7 days compared to day 1. Additionally, higher levels of insulin and glucagon were evident in NPIs seeded in VEP compared to batch matched NPIs in standard cultures at day 1 and 7 (Fig. 2E-G), confirming that the presence of the vascularized ECM is an inducer of the maturation. The insulin secretion agrees with the expression data. In fact, the insulin secretion of NPIs matured in VEP was higher than that of NPIs kept in standard culture (Fig. 3A-B). Moreover, the extension of the maturation of NPIs in VEPs up to 14 days resulted in a further improvement of the biphasic insulin secretion (Fig. 3C). The presence of endothelial cells appears relevant in this maturation process, since the seeding of NPIs in the VEP in the absence of BOEC determines a lower insulin secretion (supplementary Fig. 3). A second factor is relevant for early in vivo function. As previously reported (PMID: 30735895) the use of vascularized lung scaffold is able to improve the islet engraftment thanks to the early connection between the donor and recipient vascular network. To try to understand the relative impact of these two factors, we generated VEPs using NPIs previously fully matured in vitro by culturing for 21 days to achieve the full insulin secretion capacity (PMID: 29975241). In the absence of BOEC, once implanted in vivo, we still observed a four week delay in the time of engraftment (supplementary Fig. 5). This result confirms that the presence of endothelial cells represents a key factor in enabling the VEP's immediate engraftment and function.

4. In the NSG transplant experiments, it is not clear how the dose of NPIs in engineered VEPs was standardized to that used for transplantation at other sites.

We standardized the in vivo experiment using, where surgically possible, the same doses of NPIS (4000 NPIs) used for VEP engineering process. As originally reported in the manuscript, only the liver group received 500 NPIs and we commented this difference correlated to function in the discussion section. Now the new Fig 4A is reporting the dose of NPIs/group.

5. Discussion on the possibility of scaling up for clinical translation would be useful.

As suggested by the reviewer a dedicated comment on moving towards the clinical translation was included in the discussion

REVIEWERS' COMMENTS

Reviewer #1 (Remarks to the Author):

I have now gone through the answers from the authors to my initial critique of their manuscript and overall I think they have done a good job. However there are still a few things that need to be sorted out.

If vessels are not fenestrated, like they are within the capillaries of the pancreatic islet, how can then an efficient exchange between endocrine cells and blood take place? We should also get some more information regarding which type of nerves that are present. These two points are important for function.

Reviewer #2 (Remarks to the Author):

The authors addressed my concerns

Reviewer #1 (Remarks to the Author):

I have now gone through the answers from the authors to my initial critique of their manuscript and overall I think they have done a good job.

We thank the Reviewer for her/his positive comment.

However there are still a few things that need to be sorted out. If vessels are not fenestrated, like they are within the capillaries of the pancreatic islet, how can then an efficient exchange between endocrine cells and blood take place?

We thank the reviewer for this comment and for giving us the opportunity to clarify this point. We agree with the reviewer about the relevance of fenestrated endothelium for an efficient exchange between endocrine cells and blood in native pancreatic environment. In the context of tissue remodeling after islet transplantation or islet bio-engineering the presence of fenestrated endothelium is less mandatory for the islet function. In fact it is well described that endothelial cell lining is disrupted during islet isolation. Re-endothelialization after islet transplant under the kidney capsule is a slow and incomplete process, despite the glucose metabolism normalization in the recipient (PMID 7531955). Moreover, we need to take in consideration that VEP takes advantage of the alveolar niche to bring endothelial cells into contact with endocrine cells. We have to remember that the alveolar blood–air barrier is extremely thin (approximately 600 nm–2µm; in some places merely 200 nm) and in normal condition allow efficient exchanges even in the presence of not fenestrated endothelium. (PMID: 23606929). We agree that the islet fenestration will represent an additional step to improve the in vivo nutrient/hormonal exchange, however, based on early glycaemia normalization and long-term function (18 weeks), we assumed that hormonal exchange between endocrine cells and blood is functionally taking place independently by the presence of fenestration. A deeper and systematic study on blood micro-vessels VEP structure is ongoing and will be the subject of a future study.

We should also get some more information regarding which type of nerves that are present. These two points are important for function.

We thank the reviewer to point out this info. According to TEM acquisitions, nerve classification as “myelinated somatic peripheral nerves surrounded by perineurium” is reported in supplementary figure 7 legend.

Reviewer #2 (Remarks to the Author):

The authors addressed my concerns

We thank the Reviewer for her/his positive comment.